# Cyclical palmitoylation regulates TLR9 signalling and systemic autoimmunity in mice

Hai Ni [1,2], Yinuo Wang[3,4], Kai Yao[2], Ling Wang[2], Jiancheng Huang[2], Yongfang Xiao[2], Hongyao Chen[2], Bo Liu [3,5] ✉, Cliff Y. Yang [2,6] ✉ & Jijun Zhao [1] ✉

Toll-like receptor 9 (TLR9) recognizes self-DNA and plays intricate roles in systemic lupus erythematosus (SLE). However, the molecular mechanism regulating the endosomal TLR9 response is incompletely understood. Here, we report that palmitoyl-protein thioesterase 1 (PPT1) regulates systemic autoimmunity by removing S-palmitoylation from TLR9 in lysosomes. PPT1 promotes the secretion of IFNα by plasmacytoid dendritic cells (pDCs) and TNF by macrophages. Genetic deficiency in or chemical inhibition of PPT1 reduces anti-nuclear antibody levels and attenuates nephritis in B6.*Sle1yaa* mice. In healthy volunteers and patients with SLE, the PPT1 inhibitor, HDSF, reduces IFNα production ex vivo. Mechanistically, biochemical and mass spectrometry analyses demonstrated that TLR9 is S-palmitoylated at C258 and C265. Moreover, the protein acyltransferase, DHHC3, palmitoylates TLR9 in the Golgi, and regulates TLR9 trafficking to endosomes. Subsequent depalmitoylation by PPT1 facilitates the release of TLR9 from UNC93B1. Our results reveal a posttranslational modification cycle that controls TLR9 response and autoimmunity.

Toll-like receptors (TLRs) play essential roles in initiating innate immune responses[1–3]. Although the majority of TLRs are located on the plasma membrane, TLR3, TLR7, TLR8 and TLR9 recognize pathogenic nucleic acids in endosomes[4]. These endosomal TLRs require the chaperone protein UNC93B1 for both stability and transportation from the endoplasmic reticulum (ER)-Golgi to endolysosomes[5–8]. Once in endosomes, the ectodomains of these TLRs must be cleaved by endopeptidases before TLRs can efficiently sense nucleic acids[9–12]. Improper trafficking of endosomal TLRs results in severe autoimmunity in mice[13]. In addition to pathogen-derived nucleic acids, TLR7, TLR8, and TLR9 can recognize endogenous nucleic acids and have been associated with

autoimmune diseases such as psoriasis, systemic sclerosis, and systemic lupus erythematosus (SLE)[14–17].

In SLE, RNA sensing by TLR7 and DNA sensing by TLR9 are key to disease pathogenesis, but the underlying molecular mechanisms differ greatly[18,19]. For example, overexpression of TLR7 results in spontaneous systemic inflammation in mice, and a TLR7 gain-of-function variant has been reported in SLE patients[20–22]. However, overexpression of TLR9 does not cause lupus-like autoimmunity[18]. In addition, TLR7 deficiency ameliorated symptoms in SLE mouse models[23]. In contrast, TLR9 deficiency exacerbated SLE symptoms, suggesting that TLR9 might confer protection from this disease[18,24]. Moreover, cleaved TLR9 requires an additional release step by

[1]Department of Rheumatology and Immunology, The First Affiliated Hospital, Sun Yat-sen University, Guangzhou, Guangdong, China. [2]Department of Immunology and Microbiology, Zhongshan School of Medicine, Sun Yat-sen University, Guangzhou, Guangdong, China. [3]CAS Key Laboratory of Molecular Virology and Immunology, Shanghai Institute of Immunity and Infection, Chinese Academy of Sciences, Shanghai, China. [4]University of Chinese Academy of Sciences, Beijing, China. [5]Shanghai Huashen Institute of Microbes and Infections, Shanghai, China. [6]Key Laboratory of Tropical Disease Control (Sun Yat-sen University), Ministry of Education, Guangzhou, China. ✉e-mail: bliu@ips.ac.cn; yangkeli6@mail.sysu.edu.cn; zhjij@mail.sysu.edu.cn

UNC93B1, whereas cleaved TLR7 does not[23]. TLR9 cleavage requires an acidic environment, but TLR7 cleavage does not[25]. Nevertheless, TLR9 has recently been shown to exacerbate SLE pathogenesis via a MYD88-dependent pathway, while simultaneously protect against SLE via a MYD88-independent pathway in B cells[26]. Overall, compared to that of TLR7, the regulation of the TLR9 response seems to be much more complex.

Activated plasmacytoid dendritic cells (pDCs) express high levels of TLR7 and TLR9 and secrete large amounts of type-I and type-III interferons (IFN)[27]. This rapid cytokine response is mediated by constitutive expression of IRF7 and NF-κB in pDCs, whereas the NF-κB molecular pathway is the predominant response mediator in other immune cells, such as macrophages[28,29]. pDCs represent a unique lineage of dendritic cells that require the transcription factor E2-2 for both the development and maintenance of their identity[30]. pDCs can be activated by self-nucleic acids via endosomal TLR7 and TLR9 and produce massive amounts of IFN-I[31]. In turn, pDC-derived IFN-I can activate myeloid cells and directly regulate T and B cell differentiation[32,33]. SLE pathogenesis in humans and mouse models is characterized by excessive IFN-I signaling[34,35]. Dysregulation of IFN production in pDCs contributes to pathogenic amplification in multiple autoimmune diseases, such as type-I diabetes, psoriasis, systemic sclerosis, and SLE[14,36–39]. In mice, pDC deficiency effectively prevents the pathogenesis of systemic sclerosis and SLE[40–42]. Therefore, the aberrant production of IFN-I by pDCs is an important driver of autoimmunity.

S-Palmitoylation is a reversible posttranslational modification in which a cysteine residue is covalently attached to a saturated C16 lipid palmitate[43,44]. Palmitoylation status can contribute to protein structure and stability, vesicle trafficking and membrane anchoring[45–48]. A large family of DHHC protein acyltransferases can S-palmitoylate proteins in the Golgi, cytosol or cell membrane[49,50]. Several types of hydrolases, such as acyl protein thioesterases and palmitoyl-protein thioesterases, can depalmitoylate proteins in the cytosol or lysosomes, respectively[51–53]. In the immune system, the palmitoylation cycle has been shown to be important to several key cellular pathways and immune processes, such as the STAT3, MYD88 or NOD1/2 signaling pathways, and immune processes, such as PD-L1 degradation and antigen cross-presentation[47,54–57]. However, the importance of palmitoylation has not been shown in autoimmune diseases such as SLE.

In this study, we show that S-palmitoylation regulates the TLR9 response in pDCs and macrophages. TLR9 carries at least two S-palmitoylation sites in the ectodomain. The palmitoylation cycle begins in the Golgi, where DHHC3 palmitoylates TLR9 and ends in lysosomes, where PPT1 depalmitoylates TLR9. The TLR9 palmitoylation cycle directly regulates ligand binding and cytokine secretion by pDCs and macrophages. As a result, PPT1 genetic deficiency or PPT1 inhibitors reduce autoantibody levels and attenuate nephritis in a transgenic SLE mouse model. Palmitoylation inhibitors also repress IFNα production by human pDCs from patients with SLE. Our study outlines a posttranslational modification mechanism that regulates the TLR9 response and suggests a potential immunotherapeutic target for treating autoimmune diseases.

## Results

### PPT1 deficiency protects SLE mice from autoantibodies and nephritis

We previously studied the role of palmitoyl-protein thioesterase 1 (PPT1) in antigen presentation[57]. Since we found that PPT1-deficient classical dendritic cells type 1 (cDC1s) were much more efficient at antigen cross-presentation than wildtype cDC1s, we further explored the role of PPT1 in autoimmunity[57]. We crossed PPT1-deficient mice with B6.*Sle1yaa* mice, which carry the NZM2410/Aeg-derived SLE susceptibility 1 locus (*Sle1*) and the BXSB/MpJ-derived Yaa-containing Y chromosome (*yaa*)[15]. To our surprise, at 16 weeks, *Ppt1*[-/-] B6.*Sle1yaa*

mice exhibited a smaller spleen and fewer splenic lymphocytes than *Ppt1*[+/+] B6.*Sle1yaa* mice (Fig. 1A). We next examined a hallmark of SLE, anti-nuclear antibodies (ANAs), using a previously established ELISA method[58]. We found that *Ppt1*[-/-] B6.*Sle1yaa* mice exhibited lower serum anti-DNA antibody levels than *Ppt1*[+/+] B6.*Sle1yaa* mice (Fig. 1B). Anti-RNP/Sm antibodies were also lower at 16 weeks (Fig. 1C). The total serum IgG level was also lower in *Ppt1*[-/-] B6.*Sle1yaa* mice than it was in *Ppt1*[+/+] B6.*Sle1yaa* mice (Fig. 1D). These experiments showed that PPT1 regulated ANA levels in B6.*Sle1yaa* mice.

Next, we examined nephritis, a lethal clinical manifestation of SLE. Using H&E staining of kidney sections, we found that *Ppt1*[-/-] B6.*Sle1yaa* mice presented with a smaller glomerulus than *Ppt1*[+/+] B6.*Sle1yaa* mice (Fig.1E). Applying immunofluorescence staining, we found that *Ppt1*[-/-] B6.*Sle1yaa* mice exhibited less intense IgG and C3 staining in the nephritic glomerulus than *Ppt1*[+/+] B6.*Sle1yaa* mice (Fig. 1F). Our data suggested that PPT1 modulated nephritis in B6.*Sle1yaa* mice.

Finally, we analyzed the immune phenotype by flow cytometry. The spleens of Ppt1[-/-] B6.*Sle1yaa* mice exhibited fewer CD11b[+] myeloid cells than those of *Ppt1*[+/+] B6.*Sle1yaa* mice (Fig. 1G). *Ppt1*[-/-] B6.*Sle1yaa* mice also presented with fewer activated CD44[+] CD62L[-] CD4[+] and CD8[+] T cells (Fig. 1H, I and S1A–D). The number of MHCII[+] CXCR3[+] antibody-forming B cells was also reduced in *Ppt1*[-/-] B6.*Sle1yaa* mice compared to *Ppt1*[+/+] B6.*Sle1yaa* mice (Fig. 1J). Our results demonstrated that PPT1 controlled myeloid, effector T-cell and plasma B-cell expansion in B6.*Sle1yaa* mice.

### PPT1 inhibitor HDSF suppresses IFNα in SLE patients and SLE pathogenesis in mice

Since we showed that PPT1 may play a major role in SLE pathogenesis, we investigated the possible therapeutic value of its small molecule inhibitor HDSF[59]. First, we obtained peripheral blood from SLE patients and treated their mononuclear cells (PBMCs) with DMSO or HDSF in vitro. Consistent with the effect on mice, PPT1 inhibitors suppressed IFNα production (Fig. 2A). To examine the therapeutic potential of PPT1 in vivo, we injected HSDF into B6.*Sle1yaa* mice for 8 weeks. At the end of the treatment period, HDSF-treated B6.*Sle1yaa* mice presented with a smaller spleen and fewer splenic lymphocytes than DMSO-treated B6.*Sle1yaa* mice (Fig. 2B). Four weeks after treatment, we began to find that HDSF-treated B6.*Sle1yaa* mice showed lower serum anti-DNA antibody levels (Fig. 2C). Anti-RNP/Sm antibodies in HSDF-treated mice were also lower than that of DMSO-treated B6.*Sle1yaa* mice (Fig. 2D). Eight weeks after treatment, we observed that HDSF-treated B6.*Sle1yaa* mice exhibited a lower total serum IgG level than DMSO-treated B6.*Sle1yaa* mice (Fig. 2E). Correspondingly, HDSF-treated B6.*Sle1yaa* mice presented with a smaller glomerulus and less intense IgG and C3 staining in the nephritic glomerulus than DMSO-treated B6.*Sle1yaa* mice (Fig. 2F, G). Similarly, we detected fewer CD11b[+] myeloid cells; CD44[+] CD62L[-] CD4[+] and CD8[+] T cells; and MHCII[+] CXCR3[+] antibody forming cells (Fig. 2H–K and S2A–D). Our results demonstrated that the PPT1 inhibitor HDSF was effective in controlling autoimmunity in B6.*Sle1yaa* mice.

### PPT1 controls the cytokine response by pDCs, B cells and macrophages

Since pDC-derived IFN-I play an important role in SLE pathogenesis, we further explored the role of PPT1 in pDCs. Using enriched pDCs from healthy volunteers, we found that HDSF markedly reduced IFNα production in human pDCs (Fig. 3A). PPT1 is highly expressed in cDC1s and our previous study demonstrated that PPT1-deficiency promoted CD8[+] T cell response, which is against our current phenotype in SLE (Fig. S3A)[57]. To explain this contradiction, we began to investigate other cells heavily involved in SLE, such as macrophages, B cells and pDCs. Reanalysis of IMMGEN RNA-seq showed that the *Ppt1* transcript level was higher in macrophages (3.3-fold) and pDCs (1.85-fold) than in B cells (Fig. S3A)[60]. In particular, the *Ppt1* transcript level was upregulated

again in pDCs after CpG A stimulation (Fig. 3B). This increase occurred at the same time that the *Ifna* transcript was produced (Fig. 3B), before IFNα protein was secreted by pDCs (Fig. S3B). In order to study the function of PPT1 in pDCs, we FACS-sorted bone marrow pDCs from PPT1-deficient mice, and measured their IFNα and TNF secretion ex vivo after TLR9 or TLR7/8 stimulation. CpG A stimulation of pDCs ex vivo yielded copious amounts of IFNα, while CpG B and R848 activation yielded mostly TNF[17]. PPT1-deficient pDCs produced less IFNα and TNF than littermate wild-type controls, as measured by ELISA and intracellular staining (Fig. 3C and S3C, D). To observe the effect of PPT1 on pDC function in vivo, we infected PPT1-sufficient and PPT1-deficient mice with herpes simplex virus (HSV, Type I), a DNA virus. We observed a reduced IFNα serum concentration in PPT1-deficient mice after infection with HSV (Fig. 3D). pDCs are known to die after sufficient activation by IFN-I[61]. Correspondingly, pDC cell number was higher in PPT1-deficient mice than PPT1-sufficient mice after infection with HSV (Fig. S3E). Although pDCs are thought to be the predominant producers of IFNα in the early antiviral response, it has been difficult to

ascertain the cellular sources of serum IFNα[62]. Therefore, we generated BDCA2-DTR:*Ppt1*[+/+] and BDCA2-DTR:*Ppt1*[-/-] mixed chimeras to modulate PPT1 expression in pDCs (Fig. S3F). After HSV infection, the level of IFNα secreted by PPT1-deficient pDCs in vivo was reduced compared to that of the mixed chimera controls (Fig. S3G). Here, we showed that PPT1 regulated IFNα and TNF production by pDCs. In addition to pDCs, we also examined the role of PPT1 in B cells and macrophages. We treated B220[+] CD19[+] B cells, peritoneal cavity macrophages (Per.MΦ) and bone marrow-derived macrophages (BMDMs) from *Ppt1*[+/+]and *Ppt1*[-/-] mice with TLR9 agonists, and found that PPT1-deficient B cells secreted less IL-6 and PPT1-deficient macrophages less TNF (Fig. 3E). PPT1-deficient B cells also had less cell proliferation after CpG stimulation (Fig. S3H). Moreover, we found that PPT1 inhibitor HDSF repressed cytokine secretion by TLR9-activated pDCs, B cells and macrophages (Fig. 3F and S3I). HDSF-treated B cells also had less cell proliferation after CpG stimulation (Fig. S3J). Since HDSF almost completely abolished IFNα and TNF production, it is a possibility that HDSF-treated pDCs died ex vivo. However, after performing Annexin V

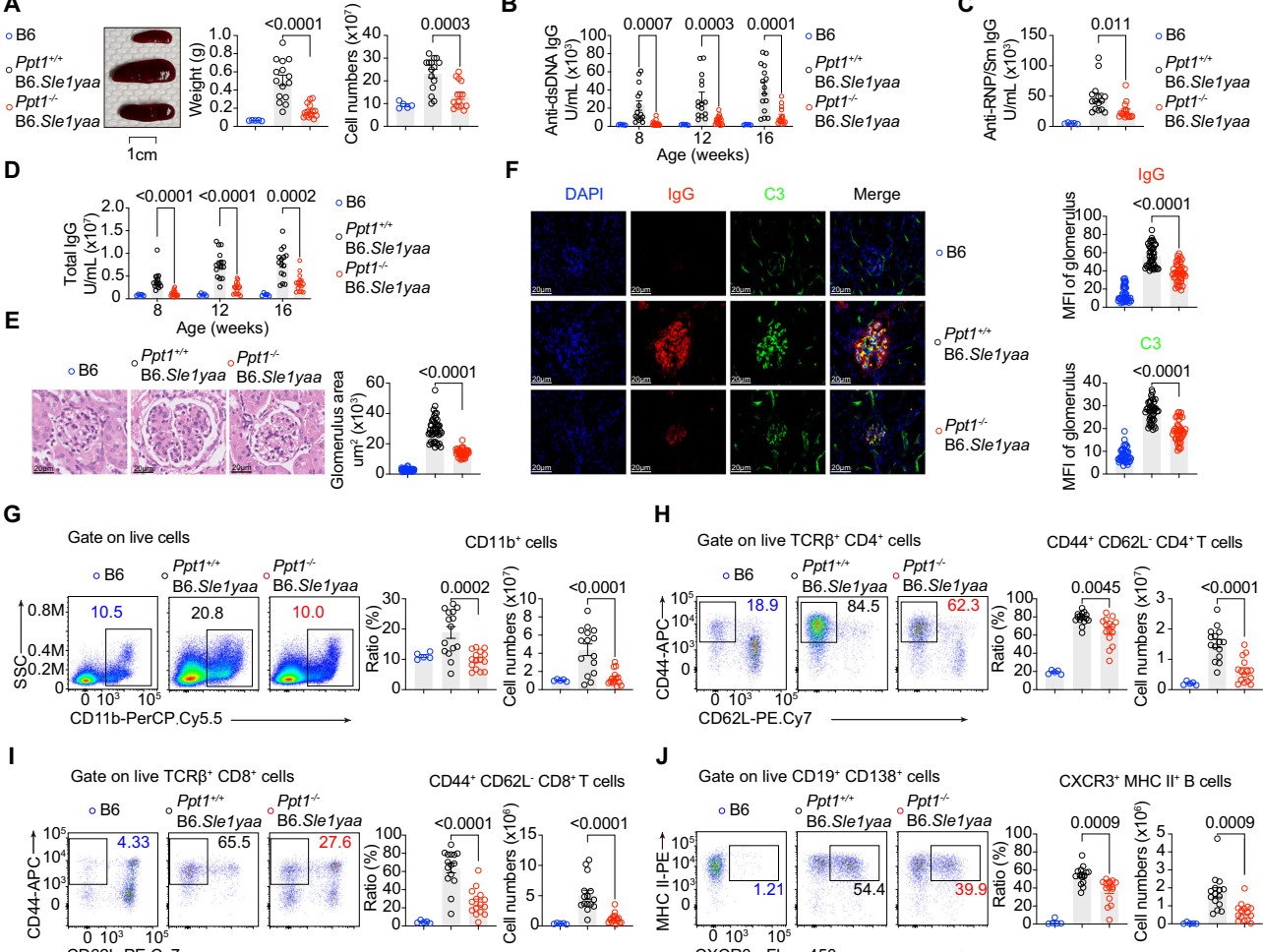

**Fig. 1 | PPT1 deficiency protects SLE mice from autoantibodies and nephritis.** C57BL/6 J (B6), *Ppt1*[+/+] B6.*Sle1yaa* and *Ppt1*[-/-] B6.*Sle1yaa* mice were sacrificed at 16 weeks (*n* = 5 mice for B6; *n* = 15 mice for *Ppt1*[+/+] B6.*Sle1yaa* and *Ppt1*[-/-] B6.*Sle1yaa* for the entire figure). **A** Spleen image (left), weight (center) and cell numbers (right) are shown. **B** Serum anti-DNA antibodies were measured by ELISAs. **C** Serum anti-RNP/Sm antibodies were measured by ELISAs at 16 weeks. **D** Serum total IgG were measured by ELISAs. **E** Glomerulus size was calculated from H&E-stained kidney sections (left). Each dot represents an individual glomerulus (*n* = 40 in B6; *n* = 44 in *Ppt1*[+/+] B6.*Sle1yaa* and *Ppt1*[-/-] B6.*Sle1yaa*). **F** Representative images from kidney immunofluorescence sections are shown (left). IgG (top) and C3 (bottom) staining

in individual glomerulus was quantified (*n* = 40 glomeruli in B6; *n* = 42 glomeruli in *Ppt1*[+/+] B6.*Sle1yaa* and *Ppt1*[-/-] B6.*Sle1yaa*). **G** Splenic CD11b[+] cell distribution is indicated by FACS plots (left), percentages (middle), and cell numbers (right). **H** Splenic CD4[+] T cell activation is indicated by FACS plots (left), percentages (middle), and cell numbers (right). **I** Splenic CD8[+] T cell activation is indicated by FACS plots (left), percentages (middle), and cell numbers (right). **J** Splenic plasma cell distribution is indicated by FACS plots (left), percentages (middle), and cell numbers (right). All data are pooled from three independent experiments (mean ± SEM.; *P* values were calculated by two-way Student's *t* test).

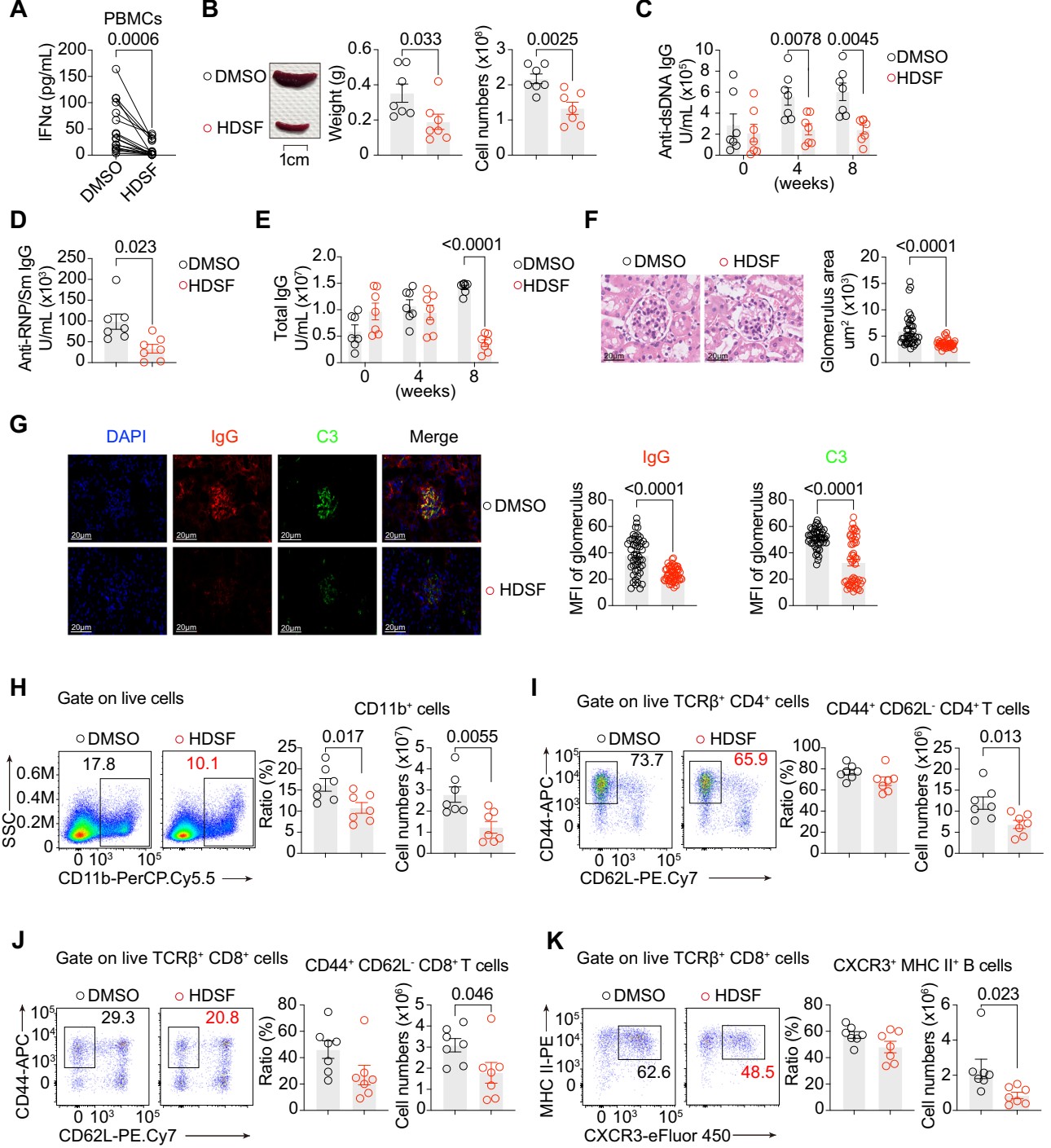

**Fig. 2 | PPT1 inhibitor HDSF suppresses IFNα in SLE patients and SLE pathogenesis in mice. A** PBMCs of SLE patients were treated with DMSO or HDSF overnight. After CpG A stimulation, IFNα levels was evaluated by ELISAs (n = 17 individuals for DMSO/HDSF). **B** 8-weeks-old B6.*Sle1yaa* mice (n = 7 mice per group for the rest of figure) were treated with DMSO or HDSF for 8 weeks before sacrifice at 16 weeks. Spleen image (left), weight (center) and cell numbers (right) are shown. **C** Serum anti-DNA antibodies were measured by ELISAs at indicated weeks of treatment. **D** Serum anti-RNP/Sm antibodies were measured by ELISAs at the end of treatment. **E** Serum total IgG were measured by ELISAs at indicated weeks of treatment. **F** Glomerulus size at 16 weeks is calculated from H&E-stained kidney sections (left). Each dot represents an individual glomerulus (n = 40 glomeruli per group). **G** Representative images from kidney immunofluorescence sections are

shown (left). IgG (middle) and C3 (right) staining in individual glomerulus was quantified (n = 50 glomerulus per group). **H** Splenic CD11b⁺ cell distribution is indicated by FACS plots (left), percentages (middle), and cell numbers (right). **I** Splenic CD4⁺ T cell activation is indicated by FACS plots (left), percentages (middle), and cell numbers (right). **J** Splenic CD8⁺ T cell activation is indicated by FACS plots (left), percentages (middle), and cell numbers (right). **K** Splenic plasma cell distribution is indicated by FACS plots (left), percentages (middle), and cell numbers (right). All data are representative of three independent experiments (mean ± SEM.; *P* values were calculated by two-way Student's *t* test), except for those presented in (**A**), which were pooled from five independent experiments (*P* values were calculated by two-tailed Wilcoxon matched-pair signed rank test).

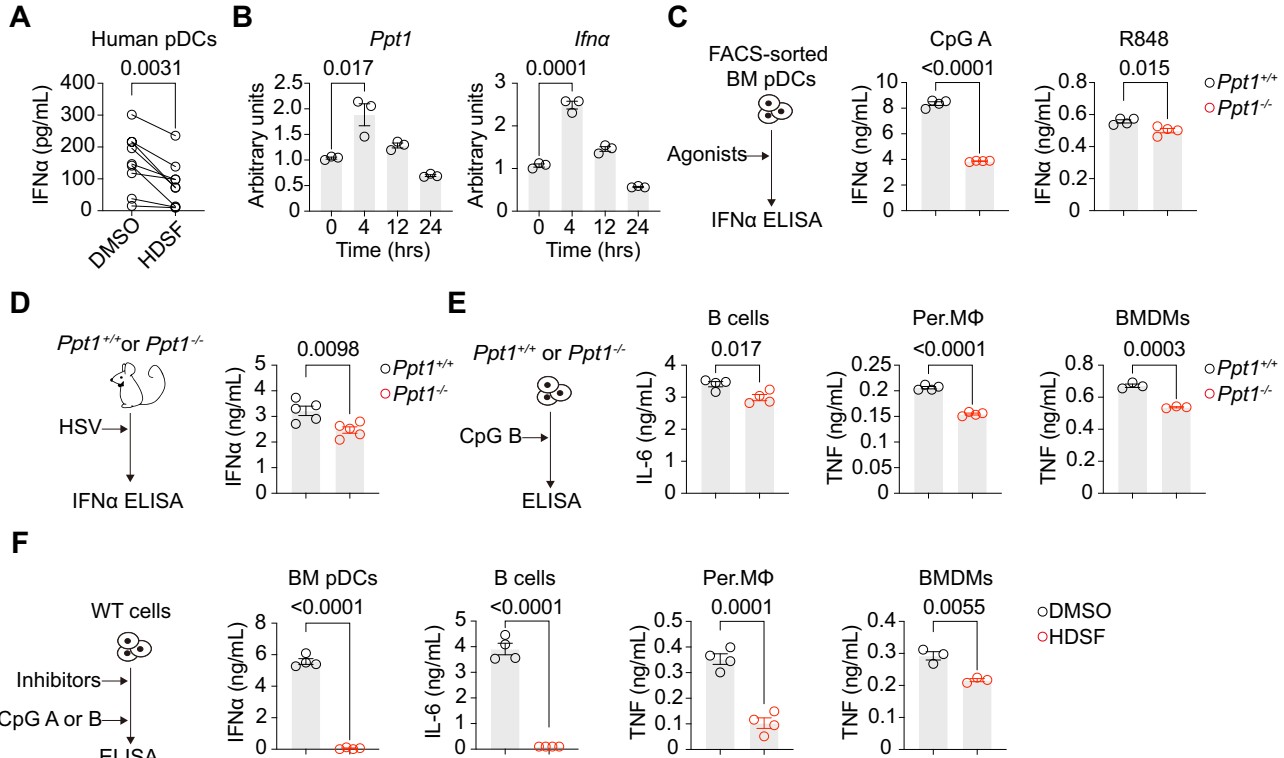

**Fig. 3 | PPT1 controls the cytokine response by pDCs, B cells and macrophages.**
**A** pDCs enriched from PBMCs of healthy volunteers were treated with DMSO or HDSF overnight. After CpG A stimulation, IFNα levels was evaluated by ELISAs ($n = 9$ individuals). **B** Sorted Flt3L-pDCs were treated with CpG A at the indicated time-points. *Ppt1* (left) and *Ifna* (right) qPCR results are shown ($n = 3$ mice per group). **C** Sorted *Ppt1+/+* or *Ppt1-/-* BM pDCs were treated with TLR9/TLR7 agonizts (left). CpG A and R848 ELISA results are shown ($n = 4$ mice per group). **D** *Ppt1+/+* or *Ppt1-/-* mice were infected with HSV (left). Serum IFNα ELISAs for HSV (right) are shown ($n = 5$ mice for HSV). **E** *Ppt1+/+* or *Ppt1-/-* indicated immune cells were treated with CpG B treatment. TNF ELISA results for CD19+ B cells, Per.MΦ and BMDMs are shown ($n = 4$

mice per group for CD19+ B cells and Per.MΦ, $n = 3$ mice per group for BMDMs). **F** WT indicated immune cells were treated with DMSO or HDSF, followed by a CpG A (pDCs) or CpG B (B cells and macrophages) pulse. TNF ELISA results for sorted BM pDCs, CD19+ B cells, Per.MΦ and BMDMs are shown ($n = 4$ mice per group for sorted BM pDCs, CD19+ B cells and Per.MΦ, $n = 3$ mice per group for BMDMs). All data are representative of three (mean ± SEM.; *P* values were calculated by two-way Student's *t* test), except for those presented in (**A**), which were pooled from three independent experiments (*P* values were calculated by two-tailed Wilcoxon matched-pair signed rank test).

staining, we found that HDSF did not affect pDC viability (Fig. S3K). In summary, we found that PPT1 also regulates cytokine production by B cells and macrophages.

## TLR9 and TLR7 are S-palmitoylated

PPT1 is an enzyme involved in depalmitoylating proteins. To investigate the possible role of palmitoylation in SLE, we analyzed several palmitoylation proteomic profile studies[63–65]. TLR9 was found to be present in one of the proteome databases[63]. Therefore, we began to explore the possibility that TLR9 might be palmitoylated. The use of click chemistry can lead to the accurate detection of all sites of palmitoylation, including S-palmitoylation and O-palmitoylation sites[66,67]. Using azido palmitate acid (also called Click-iT palmitic acid, azide) labeling, we showed that murine TLR9 (mTLR9) expressed in the RAW264.7 murine macrophage cell line was palmitoylated (Fig. 4A). Acyl-biotin exchange (ABE) is another traditional method to specifically detect S-palmitoylation[68,69]. We performed ABE and confirmed that mTLR9 was S-palmitoylated (Fig. 4B). Calnexin, a well-characterized S-palmitoylated protein, was used as a positive control, and RAB7 was used as a negative control (Fig. S4A). Both biochemical assays demonstrated that murine TLR9 was palmitoylated.

Based on a principle similar to that of ABE, we subjected mTLR9 to mass spectrometry analysis (Fig. 4C). We found that two cysteine residues, C258 and C265, were S-palmitoylated (Fig. 4D and S4B). These two amino acids are located in a small loop between the leucine rich repeat 8 (LRR8) and LRR9 in the ectodomain. However, these two

palmitoylated sites were not in the Z-loop, the actual ligand-binding groove[9]. We then proceeded to substitute these two cysteines with alanines (C258A + C265A, Mut2) and stably expressed this mutant in a TLR9-deficient RAW264.7 macrophage cell line (Fig. S4C–E). In ABE assays, we found that C258A and C265A mutations reduced the palmitoylation level of mTLR9 by 3-fold (Fig. 4E). Since TLR7 also plays an important role in SLE pathogenesis, we examined the palmitoylation status of TLR7. Using click chemistry and ABE assays, we found that mTLR7 was S-palmitoylated (Fig. 4F, G). In summary, we found that murine TLR9 and TLR7 were S-palmitoylated.

We next examined the palmitoylation status of human TLR9 (hTLR9). The ABE assays demonstrated that hTLR9 expressed in the human monocyte cell line THP-1 or 293 T cells was also S-palmitoylated (Fig. 4H, I). Here we showed that TLR9 was S-palmitoylated in human cells.

## DHHC3 palmitoylates TLR9 in the Golgi

A large family of integral membrane enzymes, known as DHHC acyltransferases, are critical for protein palmitoylation in the ER, Golgi, and plasma membrane[50,70]. To identify the DHHC proteins involved in mTLR9 palmitoylation, we screened all murine DHHC proteins (23 in total) in 293 T cell lines overexpressing mTLR9 and individual DHHC proteins via ABE assays (Fig. 5A). Among the 23 DHHC proteins we screened, we found that four proteins, DHHC3, DHHC17, DHHC18, and DHHC20, palmitoylated mTLR9 when overexpressed in vitro (Fig. 5A). All DHHC proteins carried a cysteine site in the conserved DHHC motif

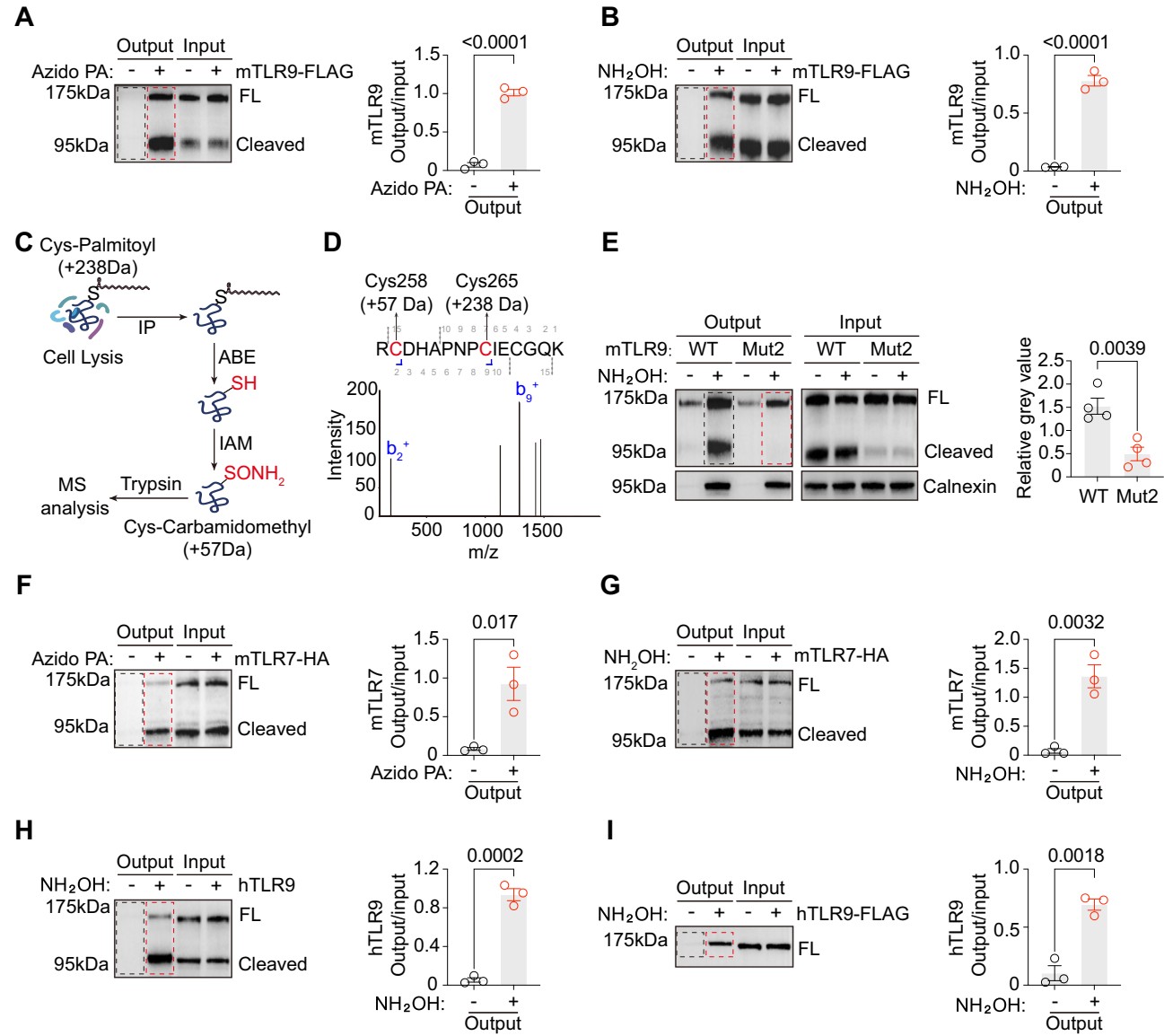

**Fig. 4 | TLR9 and TLR7 are S-palmitoylated. A** Click chemistry reactions with azido palmitic acid (PA) were performed on RAW264.7 cells transduced with mTLR9-FLAG. A representative blot (left) and mTLR9 relative palmitoylation levels (right, quantified as the output-to-input ratio of signals from all TLR9 bands, $n = 3$ replicates). **B** ABE assays were performed in RAW264.7 cells transduced with mTLR9-FLAG ($n = 3$ replicates). **C** ABE sample preparation for mass spectrometry. **D** Selected MS spectra for mTLR9 C258 and C265. **E** ABE assays were performed on *Tlr9[-/-]* RAW264.7 cells transduced with TLR9 C258A + C265A (Mut2). mTLR9 S-palmitoylation was quantified as the mTLR9 output to calnexin output ratio ($n = 4$ replicates). **F** Click chemistry reactions with azido palmitic acid (PA) were

performed on RAW264.7 cells transduced with mTLR7-HA. A representative blot (left) and mTLR7 relative palmitoylation levels (right, quantified as the output-to-input ratio of signals from all TLR7 bands, $n = 3$ replicates). **G** ABE assays were performed in RAW264.7 cells transduced with mTLR7-HA ($n = 3$ replicates). **H** ABE assays were performed in THP-1 cells ($n = 3$ replicates). **I** ABE assay of hTLR9-FLAG in 293 T cells ($n = 3$ replicates). All data are pooled from four (**E**) and three (**A**, **B**, **F**–**I**) independent experiments (mean ± SEM. *P* values were calculated by two-way Student's *t* test), except for those described in (**C**, **D**), which were obtained from one mass spectrometry run.

that was key to enzymatic activity[71–73]. Therefore, we replaced the cysteine with serine in these four potential DHHC candidates. Of these mutants, only DHHC3, DHHC17, and DHHC18 showed reduced mTLR9 palmitoylation levels, suggesting that DHHC20 may have been a false-positive hit (Fig. 5B). Since mutated DHHC3 led to the most deleterious phenotype, we knocked out DHHC3 in RAW264.7 cells and found that the mTLR9 palmitoylation rate was reduced in DHHC3-deficient cell lines (Fig. 5C and S5E). DHHC3, DHHC17 and DHHC18 had been previously found to be located in the Golgi and ubiquitously expressed in pDCs and macrophages (Fig. S5A, B)[60,74–77]. However, DHHC3 could not palmitoylate mTLR7 in vitro (Fig. 5D), suggesting that TLR7 palmitoylation process is different from that of TLR9 (Fig. 5A). Based on mTLR9 results, we also screened four human DHHC proteins with ABE

assays in 293 T cells expressing hTLR9. Only DHHC3 and DHHC17 could palmitoylate hTLR9s in vitro (Fig. 5E). Human DHHC3 and DHHC17 were also located in the Golgi, and expressed in human pDCs (Fig. S5C, D). Here we showed that Golgi-bound DHHC proteins could palmitoylate human TLR9 in vitro.

**PPT1 depalmitoylates TLR9 in lysosomes**

S-Palmitoylation is a reversible reaction[43]. We noticed that the cellular palmitoylation profile was markedly changed in bone marrow-derived pDCs (Flt3L-pDCs) and RAW264.7 cells that were stimulated with a TLR9 agonist CpG A (Fig. S6A). Using a variety of TLR agonists, we found that only the TLR9 agonists CpG A and CpG B reduced the mTLR9 palmitoylation levels in RAW264.7 cells, as shown by ABE

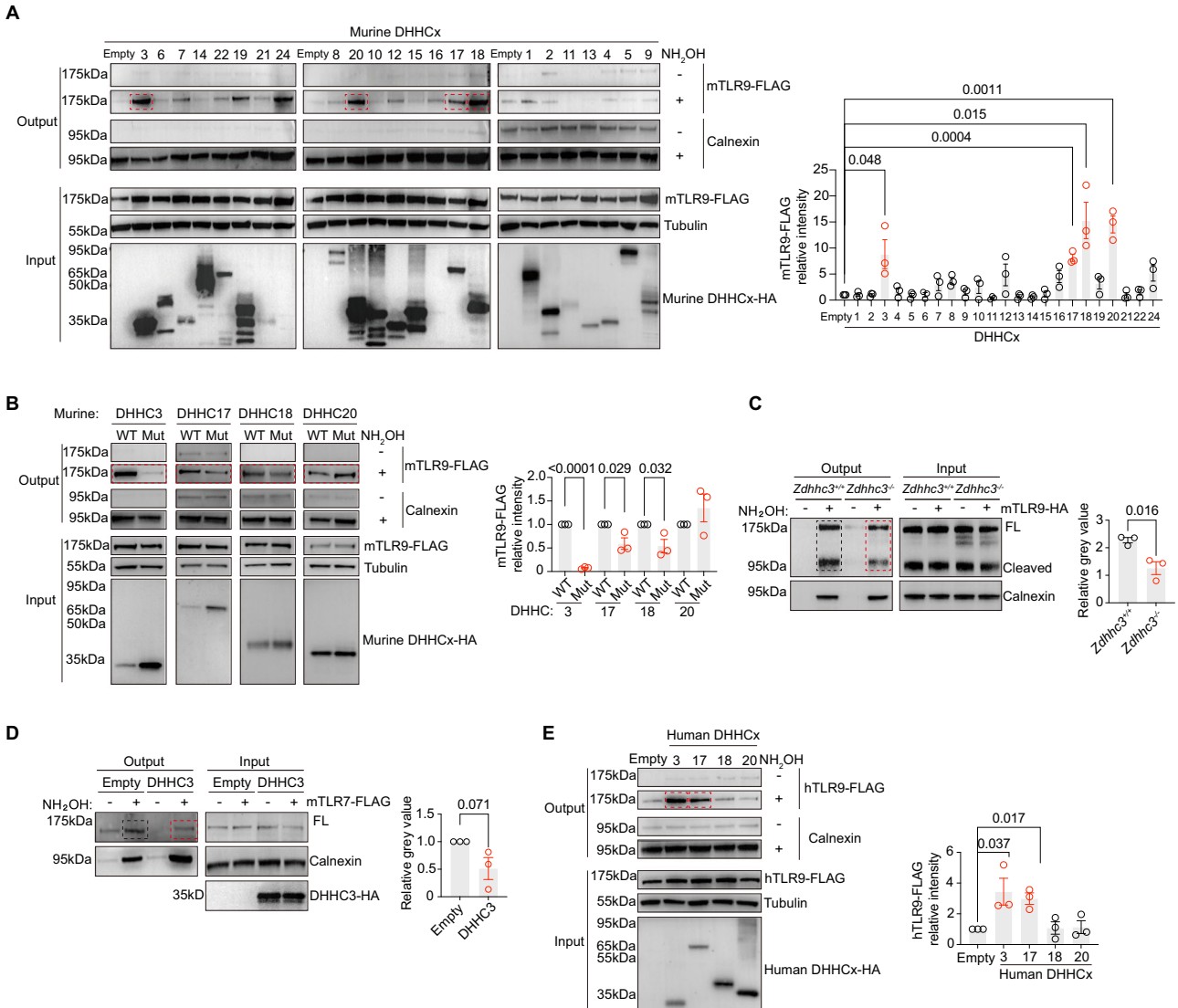

**Fig. 5 | DHHC3 palmitoylates TLR9 in the Golgi. A** ABE assays were performed in 293 T cells transfected with indicated murine DHHCs and mTLR9-FLAG. A representative blot (left) and mTLR9 relative palmitoylation levels (right, quantified as a ratio of mTLR9-FLAG signal compared to empty vector) are shown ($n = 3$ replicates). **B** ABE assays were performed in 293 T cells transfected with murine DHHC3 wild-type (WT) or C157S mutant, DHHC17 WT or C467S mutant, DHHC18 WT or C214S mutant, DHHC20 WT or C214S mutant, along with mTLR9-FLAG. A representative blot (left) and mTLR9 relative palmitoylation levels (right, quantified as a ratio of WT DHHC signal compared to respective mutants) are shown ($n = 3$ replicates) **C** TLR9 palmitoylation in DHHC3-deficient cells. ABE assays were performed on DHHC3-deficient RAW264.7 cells retrovirally transduced with mTLR9-HA. A

representative blot (left) and mTLR9 relative palmitoylation levels (right, quantified as a ratio of WT signals compared to KO) are shown ($n = 3$ replicates). **D** ABE assays were performed in 293 T cells transfected with murine DHHC3 wild-type (WT) along with mTLR7-FLAG. A representative blot (left) and mTLR7 relative palmitoylation levels (right, quantified as a ratio of mTLR7-FLAG signal compared to empty vector) are shown ($n = 3$ replicates). **E** ABE assays were performed in 293 T cells transfected with indicated human DHHCs and hTLR9-FLAG. A representative blot (left) and mTLR9 relative palmitoylation levels (right, quantified as a ratio of hTLR9-FLAG signal compared to empty vector) are shown ($n = 3$ replicates). All data are pooled from three independent experiments (mean ± SEM, $P$ values were calculated by two-way Student's $t$ test).

assays (Figs. 6A and S6B). TLR2, TLR3, TLR4 or TLR7 agonists failed to alter mTLR9 palmitoylation (Figs. 6A and S6B). In contrast, TLR7 was not depalmitoylated after R848 stimulation (Fig. 6B). In this experiment, we showed that only TLR9 S-palmitoylation, but not TLR7, was reduced after activation.

Next, we sought to identify enzymes that depalmitoylate TLR9. Hydrolases, such as acyl protein thioesterases (APTs) in the cytosol, alpha/beta-hydrolase domain (ABHD) proteins and palmitoyl-protein thioesterases (PPTs) in lysosomes, can depalmitoylate proteins[52,78,79]. Since TLR9 is trafficked from the Golgi to endosomes without being exposed its ectodomain to the cytosol, we considered only endosomal enzymes such as ABHD and PPT family members. Among the ABHD family members, ABHD10, ABHD17a, ABHD17b and ABHD17c had been

previously shown to exhibit depalmitoylation ability[80,81]. Of these candidates, only ABHD17a and ABHD17b were highly enriched in pDCs (Fig. S6C). In the PPT family, PPT1 has been shown to be a potent depalmitoylating enzyme in mice and humans, while PPT2 has been shown to exhibit specific affinity for palmitoyl-CoA but not proteins[82]. Therefore, we analyzed palmitoylation levels of mTLR9 co-expressed with ABHD17a, ABHD17b or PPT1 in 293 T cells via ABE assay. We found that only PPT1 depalmitoylated mTLR9 in vitro (Fig. 6C). In comparison, PPT1 could not depalmitoylate mTLR7 at all (Fig. 6D).

Through an ABE assay using Flt3L-pDCs from PPT1-deficient mice, we confirmed that the loss of PPT1 resulted in increased TLR9 palmitoylation in CpG A-activated Fl3tL-pDCs at 4 h and 24 h (Fig. 6E). Depalmitoylation of TLR9 after CpG A or CpG B activation was not

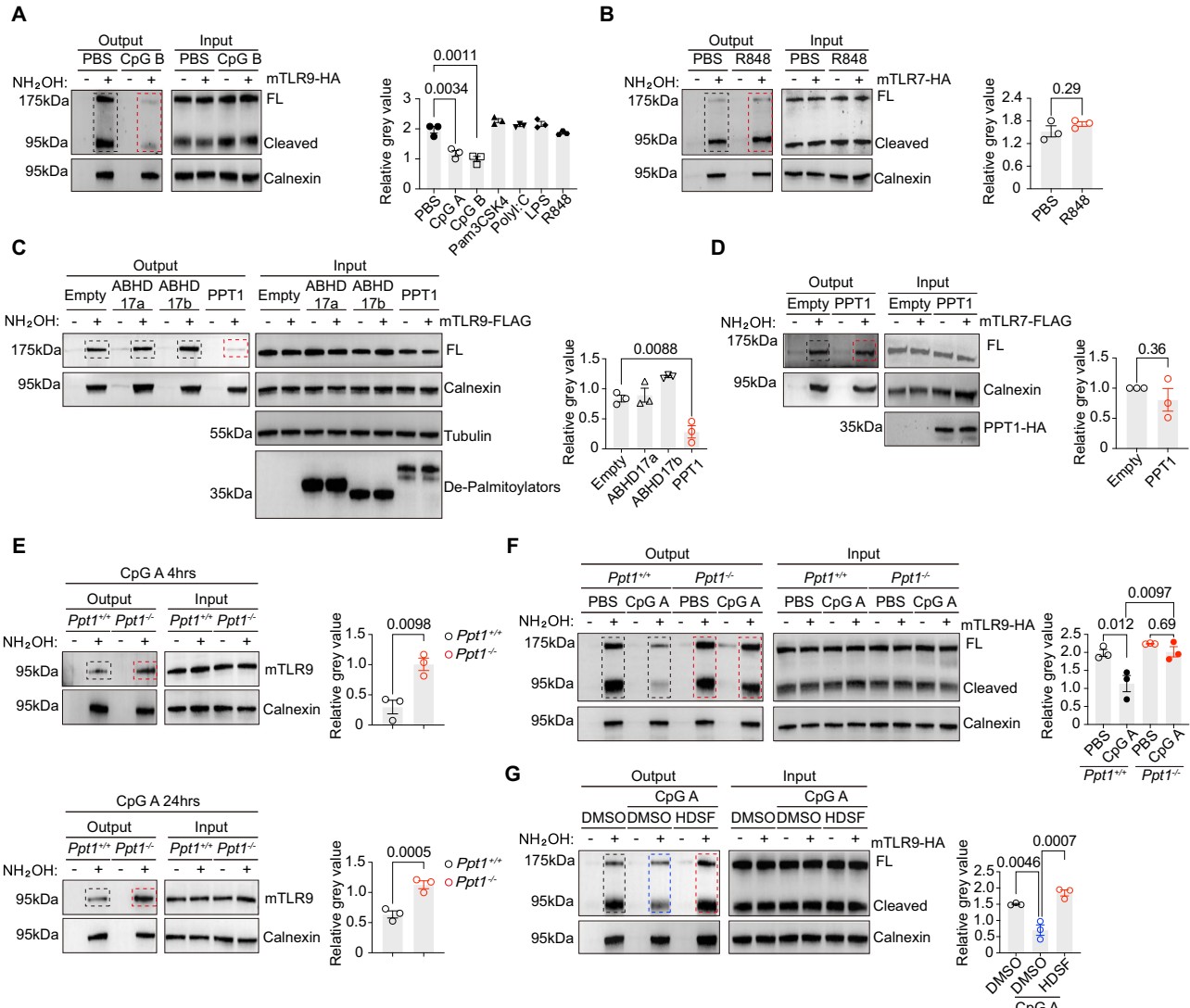

**Fig. 6 | PPT1 depalmitoylates TLR9 in lysosomes. A** ABE assays were performed on RAW264.7 cells transduced with mTLR9-HA after 4 h treatment of indicated TLR agonizts. A representative blot (left) and relative mTLR9 S-palmitoylation (right, quantified as the ratio of the mTLR9 output to the calnexin output, $n = 3$ replicates). **B** ABE assays were performed on RAW264.7 cells transduced with mTLR7-HA after 4 h treatment of R848. A representative blot (left) and relative mTLR7 S-palmitoylation (right, quantified as the ratio of the mTLR7 output to the calnexin output, $n = 3$ replicates). **C** ABE assays were performed on 293 T cells transfected with indicated depalmitoylating enzymes and mTLR9-FLAG ($n = 3$ replicates).

**D** ABE assays were performed on 293 T cells transfected with PPT1 and mTLR7-FLAG ($n = 3$ replicates). **E** ABE assays were performed on $Ppt1^{+/+}$ or $Ppt1^{-/-}$ Flt3L-pDCs after 4 h (top) and 24 h (bottom) of CpG A treatment ($n = 3$ replicates). **F** ABE assays were performed on $Ppt1^{+/+}$ or $Ppt1^{-/-}$ RAW264.7 cells transduced with mTLR9-HA after 4 h of CpG A treatment ($n = 3$ replicates). **G** RAW264.7 cells transduced with mTLR9-HA were treated with DMSO or HDSF overnight. ABE assays were performed after the addition of CpG A ($n = 3$ replicates). All data are pooled from three independent experiments (mean ± SEM. $P$ values were calculated by two-way Student's $t$ test).

evident in PPT1-deficient macrophages, suggesting that PPT1 played a role in depalmitoylation of TLR9 (Figs. 6F and S6D, E). Treatment with a general palmitoylation inhibitor 2-BP or a PPT1-specific inhibitor HDSF decreased or increased mTLR9 palmitoylation, respectively (Fig. S6F)[51,83]. We found that chemical inhibition of PPT1 was sufficient to abolish TLR9 depalmitoylation after CpG A and CpG B activation (Figs. 6G and S6G)[59]. Accordingly, 2-BP could decrease mTLR7 palmitoylation, while HDSF could not increase mTLR7 palmitoylation (Fig. S6H). Collectively, our data suggested that PPT1, a lysosomal depalmitoylating enzyme, could depalmitoylate TLR9 upon CpG activation.

### Palmitoylation regulates TLR9 trafficking and depalmitoylation regulates TLR9 release

Since TLR9 undergoes a cycle of palmitoylation and depalmitoylation, we sought to examine the impact of this cycle on TLR9 function.

Through intracellular cytokine staining, we found that the loss of two palmitoylation sites (C258A and C265A) in mTLR9 led to reduced TNF production upon CpG B stimulations (Fig. 7A). Using biotinylated CpG B in an immunoprecipitation assay, we found that these two mutations reduced the amount of cleaved mTLR9 bound to a labeled CpG ligand (Fig. 7B). Our data suggested that S-palmitoylation might regulate TLR9's binding to CpG.

Next, we examined the impact of DHHC3 and PPT1, the two key enzymes involved in the palmitoylation cycle, on mTLR9 ligand binding capability. Through ELISAs, we found that DHHC3-deficient RAW264.7 cells secreted a reduced level of TNF (Fig. 7C). Similarly, we found a reduction in TLR9-bound ligand level in DHHC3-deficient RAW264.7 cells (Fig. 7D). PPT1-deficient RAW264.7 cells also produced less TNF when stimulated with CpG B (Fig. 7E). Correspondingly, the ligand binding rate was reduced in PPT1-deficient cell lines (Fig. 7F).

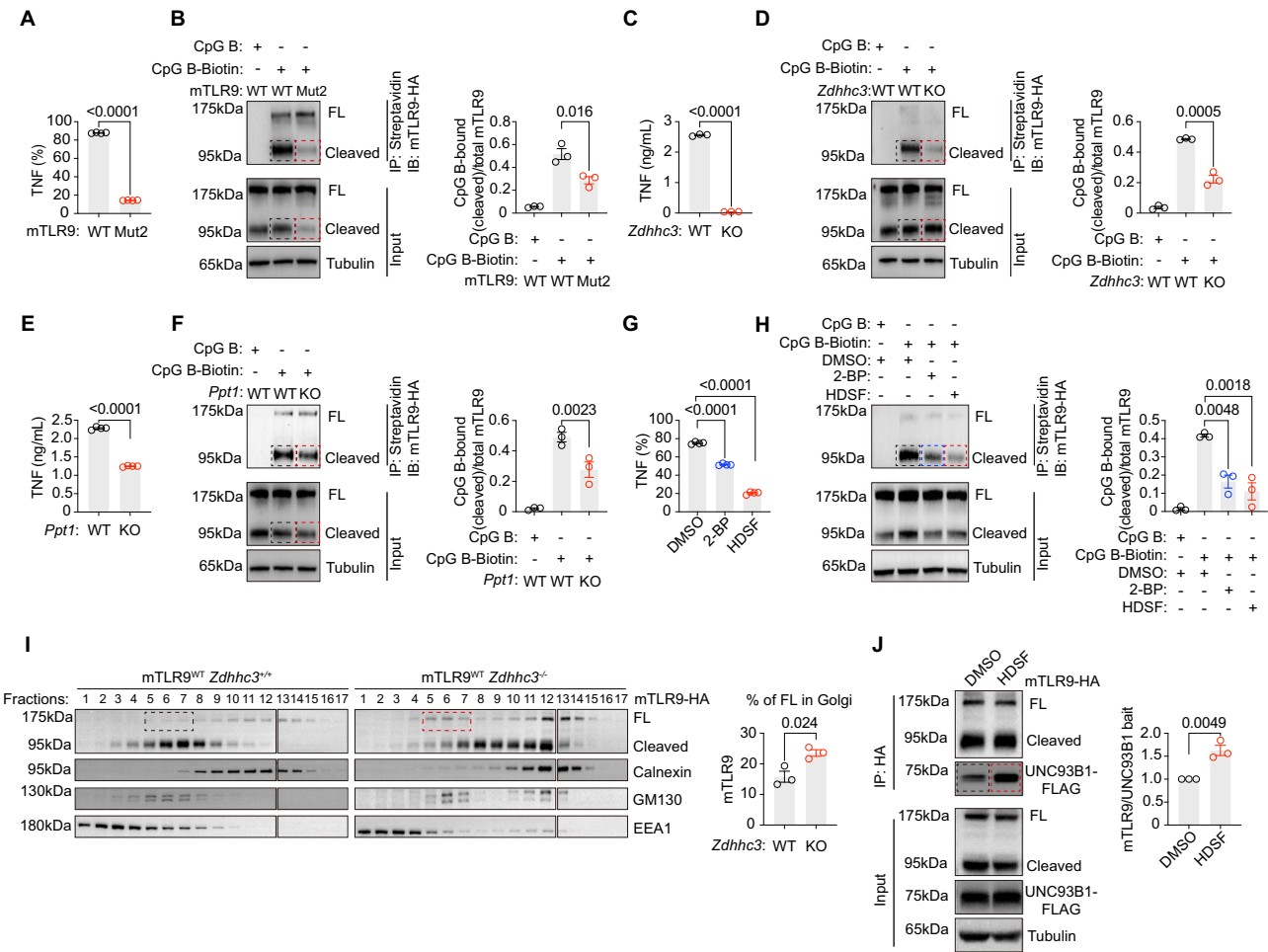

**Fig. 7 | The palmitoylation cycle regulates TLR9 ligand binding. A** TNF production was measured by intracellular staining in CpG B-treated *Tlr9*[-/-] RAW264.7 cells transduced with TLR9 Mut2 (*n* = 4 replicates). **B** Immunoprecipitation assay with CpG B-biotin were performed on *Tlr9*[-/-] RAW264.7 cells transduced with TLR9 WT or Mut2. A representative blot (left) and relative ligand binding (right, calculated as the ratio of CpG B-bound on cleaved mTLR9 to the total mTLR9 in input) are shown (*n* = 3 replicates). **C** TNF production was measured by ELISAs in CpG B-treated *Zdhhc3*[+/+] or *Zdhhc3*[-/-] RAW264.7 cells transduced with mTLR9-HA (*n* = 3 replicates). **D** Immunoprecipitation assays with CpG B-biotin were performed on *Zdhhc3*[+/+] or *Zdhhc3*[-/-] RAW264.7 cells transduced with mTLR9-HA (*n* = 3 replicates). **E** TNF production was measured by ELISAs in CpG B-treated *Ppt1*[-/-] or *Ppt1*[-/-] RAW264.7 cells transduced with mTLR9-HA (*n* = 4 replicates). **F** Immunoprecipitation assays with CpG B-biotin were performed on *Ppt1*[+/+] or *Ppt1*[-/-] RAW264.7 cells transduced with mTLR9-HA (*n* = 3 replicates). **G** RAW264.7 cells were treated with 2-BP or HDSF overnight. TNF production was measured by intracellular staining after addition of CpG B (*n* = 4 replicates).
**H** Immunoprecipitation assays with CpG B-biotin were performed on RAW264.7 cells treated with 2-BP or HDSF (*n* = 3 replicates). **I** Cell fractionation of *Zdhhc3*[+/+] or *Zdhhc3*[-/-] RAW264.7 cells transduced with mTLR9-HA showing the distributions of TLR9 in the indicated organelle markers. A representative blot (left) and the ratio of the full length mTLR9 in Golgi (right, quantified as the ratio of boxed fractions to all fractions, *n* = 3 replicates). **J** Immunoprecipitation of TLR9-HA from the DMSO or HDSF treated RAW macrophage lines in a followed by immunoblot of UNC93B1-FLAG. UNC93B1 levels in whole-cell lysates are also shown. A representative blot (left) and the ratio of the IP UNC93B1-FLAG in HDSF treated cells to the IP UNC93B1-FLAG in DMSO treated cells, *n* = 3 replicates. All data are representative of three independent experiment (mean ± SEM. *P* values were calculated by two-way Student's *t* test).

Disruption of the palmitoylation cycle by adding either 2-BP or HDSF reduced TNF production and the ligand-binding rates of TLR9 (Fig. 7G, H). Interestingly, less cleaved TLR9 was observed in mTLR9[Mut2] cells before the pulldown, whereas there was no such phenomenon in DHHC3 or PPT1-deficent cells (Fig. S7A). Our results showed that the Golgi-to-lysosome palmitoylation cycle regulated TLR9 ligand binding.

Next, we sought to differentiate the distinct function of palmitoylation and depalmitoylation. Regarding palmitoylation, we performed cell fractionation experiments, and found that more full length (FL) mTLR9 was retained in the Golgi fractions of DHHC3-deficient cells, which suggested that palmitoylation mediated by DHHC3 supported mTLR9 traffic into early endosomes where mTLR9 is cleaved to generate its functional form (Fig. 7I). However, mTLR9[Mut2] had no obvious trafficking defect (Fig. S7B). Regarding depalmitoylation in lysosomes, we performed Co-IP experiments to detect the release of mTLR9 from UNC93B1 in lysosomes, which is essential for TLR9 signal initiation. We found that HDSF reduced the amount of UNC93B1 release from mTLR9 (Fig. 7J). Our results suggest that palmitoylation regulated the trafficking of mTLR9 from the Golgi to endosomes; while depalmitoylation regulated the release of TLR9 from UNC93B1 in endolysosomes.

Lastly, general palmitoylation inhibitor 2-BP lowered IFNα secretion by PBMCs from SLE patients (Fig. S7C). 2-BP also reduced IL-6 production by B cells and their proliferation in vitro (Fig. S7D, E). However, 2-BP did not reduce the levels of anti-DNA antibodies or total IgG in B6.*Sle1yaa* mice (Fig. S7F, G). Therefore, we conclude that 2-BP regulated cytokine expression and immune cell proliferation in vitro.

## Discussion
In this study, we showed that TLR9 is palmitoylated in human and mice. Furthermore, we found that the palmitoylation and depalmitoylation of TLR9 were mediated by Golgi-resident DHHC3 and

lysosomal-resident PPT1, respectively. This palmitoylation cycle regulated cytokine production in pDCs, B cells, and macrophages. We also demonstrated the significance of the TLR9 palmitoylation cycle in a murine model of SLE in vivo.

Our results suggested that the ligand binding of TLR9 is regulated by the endosomal palmitoylation cycle. We utilized several biochemical methods, such as click chemistry, ABE assay, and mass spectrometry, to show that human and mouse TLR9 are S-palmitoylated. Click chemistry and ABE assays can only predict the general palmitoylation status of a protein. We found that TLR9 is palmitoylated at the mini-loop between LRRs. This pattern suggests that the palmitoylated amino acids may directly or indirectly facilitate the binding of CpG ligands. However, none of the potential palmitoylated sites were located in the CpG binding Z-loop[84]. Palmitoylation has been shown to be important for endosomal trafficking of membrane associated proteins, such as Rab GTPases[85]. Besides ligand binding and receptor trafficking, changes in palmitoylation level of TLR9 downstream signaling adapters, such as MYD88, might contribute to our observed phenotypes, especially in experiments using palmitoylation inhibitors[54]. Lastly, previous studies of palmitoylated proteins have consistently identified only one or two cysteine residues as S-palmitoylation sites. However, our two point mutations (Mut2) did not completely abolish TLR9 palmitoylation in ABE assays. This suggested that TLR9 may have additional palmitoylation sites. In summary, S-palmitoylation may simultaneously regulate TLR9 and TLR7 trafficking, membrane stability, and downstream signaling pathway.

Regarding the TLR9 Mut2, we have consistently noticed the weaker cleaved TLR9 bands in TLR9-Mut2 cells. However, we did not see such clearly weakened cleavage bands in our DHHC3 or PPT1-deficient cells. We also performed cellular fractionation experiments on Mut2 cells, and they had no defect in trafficking of TLR9 from Golgi to endosomes. This indicated that this Mut2 phenotype is not entirely due to the loss of palmitoylation. One possible explanation could be that these two mutations play a role in TLR9 cleavage process itself. Multiple lysosomal proteases are required for TLR9 cleavage. These two mutations may result in an altered structure, or may interfere with enzymatic process. Given that TLR9 is a highly conserved protein, these two sites may have additional functions beyond palmitoylation.

Despite our identification of DHHC3 and PPT1 as main enzymes in the endosomal palmitoylation cycle, other acyltransferases or thioesterases may be capable of palmitoylating or depalmitoylating TLR9. For example, we showed that Golgi-based DHHC17 and DHHC18 possessed the potential to palmitoylate TLR9 in vitro. To overcome the limitation of in vitro assays, experiments with mice deficient in different DHHCs may be useful for determining the function and redundancy of these enzymes in vivo. Compared to DHHC screening, only 5 lysosomal enzymes are known to depalmitoylate proteins. We screened PPT1, ABHD17a, and ABHD17b based on their high expression in pDCs, but not ABHD17c or ABHD10. Because our depalmitoylation enzyme screening was limited, we could possibly miss other proteins besides PPT1 that has a similar capacity for depalmitoylating TLR9. Additionally, it is not clear why the endosomal palmitoylation cycle must occur in the Golgi and lysosomes. The co-localization of TLR9 with DHHC3, PPT1, and other potential interacting proteins, warrants further investigation via cellular fractionation or super resolution microscopy in pDCs and macrophages. It is unclear whether specific organelles provide favorable conditions for palmitoylation or depalmitoylation of TLR9. It would be interesting to investigate which molecules signal the key lysosomal depalmitoylation step after TLR9 activation. Lastly, we have yet to confirm the human homolog of DHHC3 and PPT1 in the hTLR9 palmitoylation cycle. The endosomal locations of murine and human DHHCs are highly conserved, as indicated by human DHHC3, DHHC17, and DHHC18 being located in the Golgi. The effect of the palmitoylation inhibitor treatment of human

samples support the hypothesis that the murine and human TLR9 palmitoylation mechanisms might be similar.

Our results suggest that the TLR9 palmitoylation cycle has two major steps: palmitoylation in the Golgi, and depalmitoylation in endosomes. By performing cellular fractionation experiments, we found that DHHC3-deficient cells retained more full-length TLR9 in the Golgi, and less cleaved TLR9 was found in endosomes. Thus, palmitoylation is required for the trafficking of TLR9 from the Golgi to endosomes. Less cleaved TLR9 in the endosomes would result in a decrease of TNF secretion that was observed in DHHC3-deficient cells. Regarding the depalmitoylation step, using Co-IP experiment to probe the amount of retained TLR9 on UNC93B1, we found that there was less TLR9 bound to UNC93B1 in HDSF-treated cells, which also led to reduced TLR9 signaling. In summary, TLR9 palmitoylation regulates TLR9 trafficking from the Golgi to endosomes, and TLR9 depalmitoylation controls the release of TLR9 from UNC93B1 in endosomes. Inhibition of either step would result in less free cleaved TLR9 in the endosomes and thus reduced signaling. In accordance, HDSF-treatment and PPT1-deficiency both markedly slowed SLE pathogenesis. However, 2-BP treatment had little effect on autoantibody production in vivo. Further studies with DHHC3-deficient mice should further clarify the importance of TLR7/9 palmitoylation.

TLR7 has been shown to be an important signaling pathway for SLE pathogenesis. Using click chemistry and ABE assays, we found that TLR7 was also S-palmitoylated. However, DHHC3 overexpression could not palmitoylate TLR7, suggesting that other DHHCs might be responsible for TLR7 palmitoylation. More importantly, ABE assays showed that TLR7 was not depalmitoylated after R848 activation. In contrast, TLR9 was quickly depalmitoylated after CpG activation and stayed depalmitoylated even after 24 h. Moreover, overexpression of PPT1 do not cause TLR7 to depalmitoylate. Neither could PPT1 inhibitor HDSF change TLR7 palmitoylation level. In conclusion, TLR7 is not involved in the same cycle of palmitoylation/depalmitoylation as TLR9. Cleaved TLR7 does not need to be released from UNC93B1, while TLR9 does. The depalmitoylation of TLR9 by PPT1 facilitated its release from UNC93B1 before ligand-binding in lysosomes, while the TLR7 may not require depalmitoylation to initiate downstream signaling. Thus, it is less likely that the phenotypes we observed in the PPT1-deficient mice are mainly due to perturbed depalmitoylation of TLR7.

B cells, which are also activated by TLR7 and TLR9, arguably play a direct role in SLE pathogenesis. Although PPT1 expression in B cells was lower than pDCs and macrophages, the impact of the palmitoylation cycles on B cells should not be overlooked. Regarding the palmitoylation step, we found that 2-BP decreased IL-6 production and cell proliferation in B cells in vitro. Unfortunately, 2-BP treatment had little effect on autoantibody production in vivo. Regarding the depalmitoylaton step, we sorted B cells from Ppt1$^{+/+}$ or Ppt1$^{-/-}$ mice and examined their IL-6 production with ELISA. We found that PPT1 had an impact on IL-6 production by B cells, though the difference was very modest. Next, we examined Ppt1$^{+/+}$ or Ppt1$^{-/-}$ B cell proliferation after TLR9 stimulation. We also found that there was a slight difference. PPT1 inhibitor HSDF had a stronger effect on B cell cytokine secretion and proliferation. This pattern is consistent as in pDCs and macrophages. In conclusion, the TLR9 palmitoylation cycle had a modest impact in B cells, and may contribute partially to the phenotypes we observed in PPT1-deficient mice. Since TLR9 may be both inflammatory and suppressive in B cells, our results suggest that TLR9 palmitoylation may play a pathogenic role in B cells. It would also be intriguing to compare the differences in PPT1 expression levels in macrophages and pDCs from healthy volunteers and SLE patients, as well as changes in response to CpG A stimulation, and to detect differences and changes in TLR9 palmitoylation modification and inflammatory factor secretion.

Our previously published results regarding PPT1 in cDC1s complicates our current finding[57]. We reported that PPT1-deficient cDC1s

enhanced the priming of naive CD8[+] T cells into tissue-resident effectors and memory T cells, resulting in rapid clearance of tumors and *Listeria monocytogenes*[57]. Mechanistically, we showed that PPT1 protected steady-state DCs from viral infection by promoting antigen degradation and endosomal acidification via V-ATPase recruitment[79]. We also previously demonstrated that PPT1-deficient cDCs and MoDCs exhibit upregulated cytokine production and costimulatory molecules when activated with the TLR4 agonist LPS[57]. In the present study, we demonstrated that PPT1-deficient pDCs and macrophages under TLR9 stimulation are less activated, leading to the acquisition of a phenotype that is opposite of hyperactivated PPT1-deficient cDC1s. In PPT1-deficient mice, hyperactivated cDC1s and hypoactivated pDCs coexist, complicating the study of the overall inflammatory phenotype. In tumors and intracellular bacterial and chronic viral infection, PPT1 deficiency in cDC1s enhances the cross-presentation-specific CTL response. In SLE, PPT1 deficiency in pDCs weakens the overall T cell responses.

Originally used to treat malaria, the chloroquine derivative hydroxychloroquine (HCQ) is an effective medication to treat SLE[86]. HCQ accumulates in endosomes and prevents endosomal TLRs from binding to their ligands[87,88]. One of the possible inhibitory targets of HCQ is palmitoyl-protein thioesterase 1 (PPT1)[89]. HDSF may also exert effects by mediating the activation of the same immunosuppressive pathway as that mediated by HCQ, which also inhibits PPT1[89]. HSDF action may also inhibit MYD88 and STAT3 activation, as both proteins are palmitoylated[54,55]. In addition, it is possible that HDSF could increase endosomal pH, which may reduce the TLR9 cleavage rate dependent on acidic lysosomes[57,79]. Our results indicated a more robust inhibition of the TLR9 pathway than TLR7, which is consistent with previous findings with HCQ[90]. This further supports the notion that the most commonly used SLE therapeutic is the correct agent for inhibiting TLR9 signaling.

PPT1 may be a great drug target to treat IFN-I-mediated autoimmunity, because its inhibition may simultaneously enhance cDC1-dependent cross-priming and suppress pDC-dependent IFN-I production. Many immunosuppressive drugs have the same drawback: they have been associated with frequent opportunistic infections and tumorigenesis due to weakened immunity in patients. In this study, we demonstrated that PPT1 inhibitor HDSF effectively controls IFN-I production in mice and humans. We previously showed that PPT1-deficient mice had a lower tumor burden and a higher survival rate than PPT1-sufficient mice in xenograft tumor models, possibly due to 4-fold expansion of tissue-resident memory CD8[+] T cells[57]. PPT1-deficient mice also had 30-fold lower bacterial load when infected with *Listeria monocytogenes*, an intracellular bacterium, due to 3-fold expansion of KLRG-1[+] short-lived effector CD8[+] T cells[57]. In addition, PPT1-deficient mice had 3-fold lower viral titers when infected with lymphocytic choriomeningitis virus (LCMV) clone 13, a chronic infection variant[57]. Similarly, adoptive transfer of WT BMDCs treated with HDSF was able to extend mice survival in xenograft tumor transplantation models and lower bacterial load by 50-fold in *Listeria monocytogenes* infection models[57]. PPT1 inhibitors such as HDSF may be the first drug in its class to suppress autoimmunity while enhancing immunity against tumorigenesis and infection.

In conclusion, our data reveal a Golgi-to-lysosomal palmitoylation pathway that regulates the TLR9 and TLR7 response in mice and humans. We showed that disruption of this endosomal palmitoylation/depalmitoylation cycle suppressed TLR9 ligand binding and cytokine production in pDCs and macrophages. PPT1 genetic deficiency or chemical inhibition suppressed anti-DNA autoantibodies and attenuated nephritis in B6.*Sle1yaa* mice. Given its opposite action in cDC1s and pDCs, PPT1 may be a potential pharmaceutical inhibitory target for SLE treatment, due to possible added benefit of enhancing the CTL response against cancer and pathogens.

## Methods

### Transgenic mice

All mice were bred and maintained in specific pathogen-free conditions at GemPharmatech Co., Ltd., and BSL3 facilities at Sun Yat-sen University, according to the institutional guidelines and protocols approved by the Animal Ethics Committee of Sun Yat-sen University, Guangzhou, Guangdong, China. C57BL/6 J (WT), B6.SJL-*Ptprc*ᵃ*Pepc*ᵇ/BoyJ (Strain #: 002014, CD45.1[+]), B6.129S6-*Ppt1*ᵗᵐ¹ᴴᵒᶠ/SopJ, (Strain #: 006566, *Ppt1*[−/−])[91], B6. Cg-*Sle1*ᴺᶻᴹ²⁴¹⁰/ᴬᵉᵍ*Yaa*/DcrJ (Strain #: 021569, B6.*Sle1yaa*)[15], and C57BL/6-Tg (CLEC4C-HBEGF)956Cln/J (Strain #: 014176, BDCA2-DTR)[62] mice were purchased from the Jackson Laboratory. B6.129S6-Ppt1tm1Hof/SopJ (*Ppt1*[−/−]) mice were back-crossed for ≥10 generations with C57BL/6 J. *Ppt1*[−/−] B6.*Sle1yaa*, and *Ppt1*[+/+] B6.*Sle1yaa* mice were generated by crossing B6.Cg-*Sle1*ᴺᶻᴹ²⁴¹⁰/ᴬᵉᵍ*Yaa*/DcrJ (B6.*Sle1yaa*) males with backcrossed B6.129S6-Ppt1tm1Hof/SopJ (*Ppt1*[−/−]) females for two generations. Age- and sex-matched mice at 6-16 weeks of age were used in this study.

### Generation of chimeras

The chimeras were generated as previously described[62]. To establish BDCA2-DTR: *Ppt1*[+/+] or BDCA2-DTR: *Ppt1*[−/−] mixed chimeras, CD45.1[+] mice were lethally irradiated and then injected i.v. with $5 \times 10^6$ bone marrow cells harvested from *Ppt1*[+/+] or *Ppt1*[−/−] littermates and $5 \times 10^6$ bone marrow cells from age- and sex-matched BDCA2-DTR mice. All chimeras were fed antibiotics for 2 weeks and then rested for at least an additional 8 weeks to allow reconstitution of immune cells.

### Murine SLE model

Measurement of anti-dsDNA antibodies in serum was previously described[32]. Briefly, 96-well flat bottom ELISA plates were coated with 50 μl of 50 μg/ml poly-L-lysine (Sigma, P8920) diluted in 1x PBS overnight at 4 °C. The plates were washed with 1x PBS/0.05% Tween, coated with 10 μg/ml dsDNA (Invitrogen, 15633019) diluted in PBS and incubated overnight at 4 °C. Excess antigen was washed off using 1x PBS/0.05% Tween. Antigen-coated plates were blocked with 250 ml/well of 1x PBS/1%BSA for at least 2 h at RT. Blocked plates were washed once with 1x PBS/0.05% Tween, coated with diluted serum in PBS and incubated overnight at 4 °C. Unbound serum antibodies were washed off the plates with 1x PBS/0.05% Tween. Bound IgG was detected using a 1:1000 dilution of goat AP-conjugated anti-mouse IgG (Invitrogen, G-21060) diluted in 1x PBS/1% BSA for 2 h. Unbound secondary antibody was washed off the plates with 1x PBS/0.05% Tween and developed using diethanolamine substrate buffer (Thermo Fisher Scientific, 34064) and PNPP phosphatase substrate tablets (Sigma, SIGMAS0942). For relative quantitation of antigen-specific IgG titers, serum from an anti-DNA-positive animal (B6.*Sle1yaa* > 16 weeks old) was used as the standard after serial double dilution. The O.D. at the lowest serum dilution was arbitrarily assigned a value of 100 μ/ml. The standard curve was plotted as O.D. at 405 nm versus antigen concentration (μ/ml).

Total serum IgG titers were measured by coating ELISA plates with IgG-positive serum and adding samples after serial double dilution. The bound serum antibody was detected with goat AP-conjugated anti-mouse IgG (Invitrogen, G-21060).

Serum anti-RNP/Sm antibodies titers were measured by RNP/Sm IgG ELISA kit (Calbiotech, RN038G-KIT) according to the manufacturer's instructions.

For kidney H&E analysis, mouse kidneys were fixed in formalin and embedded in paraffin, and tissue sections were stained with H&E. Glomerular size was analyzed in a blinded manner using ImageJ software (Version: 2.1.0/1.53c). Glomerular size was determined from an area with at least 10 glomeruli per kidney section from at least 3 mice in each group.

For kidney immunofluorescence analysis, mouse kidneys were frozen in OCT (Tissue Tek). Five- to ten-micrometer-thick kidney

sections were fixed using chilled acetone for 10 min. Sections were stained with DAPI, rabbit anti-mouse IgG (Thermo Fisher Scientific, A27022) and rat anti-mouse C3 (Abcam, ab11862). Images were captured using a Keyence BZ-X710 fluorescence microscope at 40X magnification. Immunofluorescence intensity was quantitated in a blinded manner using ImageJ (Version: 2.1.0/1.53c). Immunofluorescence intensity of at least 10 glomeruli per kidney section from 3 or 4 mice in each group was measured.

### Viral infections

The viral infections were previously described[92]. For HSV-1 infection, $5 \times 10^6$ PFU of virus was diluted in PBS and injected i.v. into $Ppt1^{+/+}$ and $Ppt1^{-/-}$ mice. Serum was collected at 5 h post infection. 200 ng/mouse of diphtheria toxin (Sigma, D0564) was injected i.p. into BDCA2-DTR:$Ppt1^{+/+}$ and BDCA2-DTR:$Ppt1^{-/-}$ mixed chimeras 24 h before infection with HSV-1. Serum was collected at 5 h post infection. Serum was stored at −80 °C until analyzed.

### Palmitoylation inhibitors

For in vitro palmitoylation inhibitor treatment, cells were pretreated overnight at 37 °C with 50 μM 2-BP (Sigma, 21,604), 10 μg/mL HDSF (APExBio, C5608) or DMSO as the control. For in vivo treatments, palmitoylation inhibitor treatment was initiated in 8-week-old B6.$Sle1yaa$ mice. Dissolved HDSF or DMSO was diluted in saline and intravenously injected at 20 mg/kg every other day. For 2-BP injection, dissolved 2-BP or DMSO was diluted in saline and intraperitoneally injected at 40 mg/kg every other day.

### Palmitoylation assays

Click chemistry assays were previously described[67]. Cells were treated with 100 μM azido palmitic acid (Click-iT palmitic acid-azide) (Thermo Fisher Scientific, C10265) in complete DMEM and then incubated at 37 °C under 5% $CO_2$ for 6 h. After incubation, the medium was removed, and the cells were washed three times with phosphate-buffered saline before the addition of lysis buffer containing a protease inhibitor cocktail (Thermo Fisher Scientific, 78443). Insoluble material was eliminated from the protein-containing cell lysates by centrifugation. Protein concentrations were measured using the BCA assay (Beyotime, P0012). 100 μg of protein was transferred to a fresh tube to serve as 'input'. 200 μg of azide-labeled protein sample was reacted with biotin-alkyne using a Click-iT Protein Reaction Buffer Kit (Thermo Fisher Scientific, C10276) following protocols on the manufacturer's instruction sheet. Then, the biotin alkyne-azide-palmitic-protein complex was pulled down by streptavidin-agarose beads (Thermo Fisher Scientific, 20361). Proteins after enrichment were labeled as "output". After washing, bead-bound samples were resuspended in 0.2 ml of 2× reducing SDS-PAGE sample buffer and boiled for 10 min at 95 °C.

Acyl-biotin exchange (ABE) assays were previously described[69]. Samples were suspended in 1 ml of lysis buffer (Thermo Fisher Scientific, 87787) containing a protease inhibitor cocktail (Thermo Fisher Scientific, 78443) and 50 mM N-ethylmaleimide (Sigma, E3876) and incubated overnight at 4 °C. Samples were precipitated with chloroform/methanol, briefly air-dried, and resuspended in 1 mL of resuspension buffer (50 mM Tris-HCl, pH 7.2; 2% SDS; 8 M urea; and 5 mM EDTA). Samples were then divided into 2 solutions; 1 aliquot was combined with 0.5 mL 1 M hydroxylamine (Macklin, H811237), and the negative control was combined with 0.5 mL 1 M NaCl. Samples were incubated at RT for 1 h, and then proteins were precipitated by chloroform-methanol treatment. Protein pellets were resuspended in 1 mL of resuspension buffer containing 10 μM EZ-Link HPDP-Biotin (Thermo Fisher Scientific, 21341) and incubated at RT for 2 h. Excess biotin was removed via chloroform-methanol precipitation. Protein pellets were dissolved in 0.25 mL of resuspension buffer. Protein concentrations were measured by BCA assay. 100 μg of protein was

transferred to a fresh tube to serve as 'input'. 800 μg of protein in solution was diluted 1:10 with PBS and mixed with 20 μL of streptavidin-agarose beads (Thermo Fisher Scientific, 20361) for 1 h. The beads were washed 3 times with PBS containing 1% SDS. Bead-bound samples were labeled as "output", and were resuspended in 0.2 ml of 2× reducing SDS-PAGE sample buffer (Sigma, S3401). The samples were boiled for 10 min at 95 °C before SDS-PAGE.

### Mass spectrometry

Sample preparation with ABE was previously described[68,69]. RAW264.7 cells carrying mTLR9-Flag were lysed, solubilized and immunoprecipitated with anti-FLAG affinity beads (Sigma, A2220). After washing with 1 ml of lysis buffer three times at 4 °C, the beads were subjected to the ABE assay preparation procedure and an alkylation reaction using an IP-ABE Palmitoylation Kit for MS (Aims, AM10417) following the protocols on the manufacturer's instruction sheet. The eluted sample was used for further MS analysis.

Samples were digested into peptides by trypsin, and then analyzed by liquid chromatography-tandem mass spectrometry using a nanoflow UPLC system (Thermo Fisher Scientific) and Q Exactive Hybrid Quadrupole-Orbitrap mass spectrometer (Thermo Fisher Scientific). Peptide mixtures were separated on an Easy-nLC 1200 system on a C18 nanocolumn (3 μm, 75 μm × 15 cm, made in house) at a flow rate of 600 nL/min. A 66-min linear gradient was established as follows: 4% B to 8% B in 2 min, 8% B to 28% B in 43 min, 28% B to 40% B in 10 min, 40% B to 95% B in 1 min and held for 10 min at 95% B. For data acquisition, a top 20 scan mode with an MS scan range of m/z 300–1800 was used. To better identify modified amino acid sites, each precursor ion was fragmented with HCD. The collision energy of the HCD was set to 28, and the ETD reaction time was set automatically according to the m/z and ion charge state of each precursor. The raw MS files were analyzed and searched against a target protein database based on the sample species using Byonic (Version 4.3). The parameters were set as follows: the protein modifications were carbamidomethylation (C) (+57.0215), acetyl (protein N-term) (variable), oxidation (M) (variable), and palmitoylation (C, K, S, T, protein N-term) (+238.230); the enzyme specificity was set to trypsin; the maximum number of missed cleavages was set to 3; the precursor ion mass tolerance was set to 20 ppm, and the MS/MS tolerance was 20 ppm. Only peptides identified with high confidence were chosen for downstream protein identification analysis.

### Ligand binding

Ligand binding immunoprecipitation methods were previously prescribed[93]. Cells were fed 1 μM CpG B-biotin (InvivoGen, tlrl-1826b) for 4 h. Then, the medium was removed, and the cells were washed three times with phosphate-buffered saline followed by lysis buffer containing a protease inhibitor cocktail. Insoluble materials were eliminated from the protein-containing cell lysates by centrifugation. Protein concentrations were measured by BCA assay. Protein from the lysates (1 mg per 1 mL) were suspended in 1 ml of lysis buffer containing a protease inhibitor cocktail and incubated with 20 μL of streptavidin-agarose beads overnight at 4 °C with end-over-end rotation. Unbound proteins were removed by washing 4 times with 1 ml of lysis buffer. Bead-bound proteins were denatured in 2× reducing SDS-PAGE sample buffer and boiled for 5 min at 95 °C.

### Human samples

This study was approved by the Ethics Committee of First Affiliated Hospital, Sun Yat-sen University (Approval No. 2022-436). Informed consent was obtained from all subjects. All patients met the American College of Rheumatology classification criteria for systemic lupus erythematosus[94]. Patients with concurrent infection, malignancy or other autoimmune diseases were excluded from the study. There was no exclusion criterion for healthy volunteers.

Peripheral blood mononuclear cells (PBMCs) were isolated from whole blood by using a Hisep LSM (Ficoll). pDCs were enriched from freshly isolated PBMCs (purity >50%) by negative selection using human Plasmacytoid Dendritic Cell Isolation Kit II (Miltenyi, 130-097-415), according to the manufacturer's instructions. Cells were cultured in complete RPMI-1640. For cytokine production, cells were pre-incubated with DMSO, 2-BP or HDSF and then stimulated with 1 μM CpG A (InvivoGen, tlrl-2216-1). Cytokine production was measured by ELISA.

## Primary cell culture

To establish Fl3tL-pDCs, $3 \times 10^6$ bone marrow cells per well were cultured in tissue culture-treated 6-well plates in 3 ml of complete RPMI-1640 (RPMI-1640 supplemented with 10% FCS, 1% L-glutamate, 1% sodium pyruvate, 1% MEM with nonessential amino acids, 1% penicillin/streptomycin, and 55 μM 2-mercaptoethanol). Fl3tL-B16 culture supernatant was added as conditioned medium to account for 20% of the medium and provide Flt3L to cells. Fl3tL-B16 (ATCC, CRL-6475) cells were retrovirally transduced in house with murine Flt3l plasmid. Flt3L-pDCs were gated as live$^+$ CD11c$^+$ B220$^+$ SiglecH$^+$ cells. These cells were harvested on Day 7.

To establish BMDMs, $5 \times 10^6$ bone marrow cells per well were cultured in 90-mm nonpyrogenic sterilized petri dishes in complete DMEM (DMEM supplemented with 10% FCS, 1% L-glutamine, 1% sodium pyruvate, 1% MEM with nonessential amino acids, 1% penicillin/streptomycin, and 55 μM 2-mercaptoethanol) and L929 mouse fibroblast (ATCC, CRL-6364TM) culture supernatant was added as conditioned medium to account for 30% of the medium and provide macrophage colony stimulating factor (M-CSF). One-half of the medium was replenished on Day 3 and Day 6, and the cells were harvested on Day 7. To harvest the macrophages, the cultured cells were washed once with ice-cold PBS and then incubated in PBS containing 20 mM EDTA and 20% FCS at 37 °C for 5 min. BMDMs were gated as live$^+$ CD45$^+$ CD11b$^+$ F4/80$^+$ cells.

## Immune cell isolation

To harvest immune cells from lymphoid tissue, organs were minced, ground, and passed through a 70-μm nylon mesh. Erythrocytes were removed using ammonium-chloride-potassium lysis buffer (150 mM ammonium chloride, 10 mM potassium bicarbonate, and 0.1 mM EDTA). The cells were counted using a Beckman Coulter CytoFLEX flow cytometer. Before cell sorting, pDCs were negatively enriched by the addition of the following cocktail of biotinylated antibodies to the culture: anti-IgG, anti-IgD, anti-CD19, anti-CD93, anti-CD5, anti-Ly6G, anti-Ter119, anti-CD41, anti-NK1.1, anti-TCRb, anti-CD3, anti-CD11b, anti-CD24 and anti-F4/80 antibodies at a 1:1000 dilution for 20 min at 4 °C. Unlabeled cells were collected, and pDCs were sorted as B220$^{lo}$ CD11c$^{lo}$ SiglecH$^+$ BST-2$^+$ (purity >95%). Cell-sorting experiments were conducted on a BD FACSAria III flow cytometer. Staining was performed at 4 °C in the presence of an Fc blocker (BD, Clone 2.4G2) and FACS buffer (PBS, 0.5% BSA, 2 mM EDTA, and 0.1% sodium azide). Peritoneal cavity macrophages (Per. MΦ) were obtained by peritoneal cavity lavage twice with 5 mL of ice-cold PBS. Per. MΦ were collected in a 15-ml tube and centrifuged at $500 \times g$ for 5 min. After short incubation on tissue-culture plates and washing of unattached cells, Per. MΦ were considered purified (purity >90%).

A table with details of the biotinylated antibodies is presented below:

Invitrogen IgG Catalog #: 13-4013-85
Invitrogen IgD clone: 11-26 Catalog #: 13-5993-82
Invitrogen CD93 clone: AA4.1 Catalog #: 13-5892-82
Invitrogen CD5 clone: 53-7.3 Catalog #: 13-0051-82
Invitrogen CD49b clone: DX5 Catalog #: 13-5971-85
Invitrogen TCR β clone: H57-597 Catalog #: 13-5961-85
Invitrogen CD11b clone: M1170 Catalog #: 13-0112-82
Invitrogen CD24 clone: M1/69 Catalog #: 13-0242-82

Invitrogen F4/80 clone: BM8 Catalog #: 13-4801-81
BioLegend Ly6G/Ly6C (Gr-1) clone: R86-8C5 Catalog #: 108404
BioLegend CD41 clone: MWReg30 Catalog #: 133930
TONBO CD19 clone: 1D3 Catalog #: 30-0193-U500
TONBO TER-119 clone: TER-119 Catalog #: 30-5921-U500
TONBO CD3 clone: 145-2C11 Catalog #: 30-0031-U500
All antibodies were used at a dilution of 1:1000.

## Flow cytometry

Flow cytometry was performed on a Beckman Coulter CytoFLEX and analyzed using FlowJo software (Tree Star, version X). MFI was calculated on the basis of the genomic mean with FlowJo software. Staining was performed at 4 °C in the presence of an Fc blocker (Clone 2.4G2; BD) and FACS buffer (PBS, 0.5% BSA, 2 mM EDTA, and 0.1% sodium azide). For intracellular cytokine staining, 10 μg/ml brefeldin A (BFA, eBioscience) was added to TLR ligands with the following factors at the following concentrations: 1 μg/ml R848 (InvivoGen, tlrl-r848), 1 μM CpG A (InvivoGen, tlrl-1585), 1 μM CpG B (InvivoGen, tlrl-1668), 1 μg/mL Pam3CSK4 (InvivoGen, tlrl-pms), 100 ng/mL LPS (Sigma, L2654-1MG), and 500 ng/mL poly(I:C) (InvivoGen, tlrl-pic) for 4–16 h before staining using an intracellular staining kit (Thermo Fisher Scientific, 88-8824-00). The number of apoptotic cells was measured using an Annexin V-FITC Apoptosis Detection Kit (Invitrogen, BMS500FI) according to the manufacturer's instructions.

A detailed table of the antibodies used for flow cytometry is presented below.

The following antibodies were used at a dilution of 1:400 (unless otherwise indicated):

Invitrogen CD19 eFluor 450 Clone: 1D3 Catalog #: 48-0193-82
Invitrogen CD3e APC clone: 17A2 Catalog #: 17-0032-82
Invitrogen CD11b PerCP-Cyanine5.5 clone: M1/70 Catalog #: 45-0112-82
Invitrogen IgD APC-Cyanine7 clone: 11-26c Catalog #: 47-5993-82
Invitrogen IgM PE clone: II/41 Catalog #: 12-5790-82
Invitrogen B220 FITC clone: RA3-6B2 Catalog #: 11-0452-82
Invitrogen CD38 PE-Cyanine7 clone: 90 Catalog #: 25-0381-82
Invitrogen GL7 eFluor 450 clone: GL-7 Catalog #: 48-5092-82
Invitrogen CD1d PE clone: 1B1 Catalog #: 12-0011-82
Invitrogen CD21/CD35 APC-Cyanine7 clone: 8D9 Catalog #: 47-0211-82
Invitrogen TCR β PE clone: H57-597 Catalog #: 12-5961-82
Invitrogen CD69 eFluor 450 clone: H1.2F3 Catalog #: 48-0691-82
Invitrogen CD62L PE-Cyanine7 clone: MEL-14 Catalog #: 25-0621-82
Invitrogen BST2 FITC clone: eBio927 Catalog #: 11-3172-82
Invitrogen SIGLEC-H APC clone: eBio440c Catalog #: 17-0333-82
Invitrogen TNF PE-Cyanine7 clone: MP6-XT22 Catalog #: 25-7321-82
Invitrogen CD45.1 PE-Cyanine7 Clone: A20 Catalog #: 25-0453-82
Invitrogen CD45.2 APC-Cyanine7 Clone: 104 Catalog #: 47-0454-82
BioLegend CD44 APC-Cyanine7 clone: IM7 Catalog #: 103028
BioLegend CD44 APC clone: IM7 Catalog #: 103012
BioLegend B220 PE clone: RA3-6B2 Catalog #: 103208
BioLegend PD-1 PE-Cyanine7 clone: 29 F.1A12 Catalog #: 135216
BioLegend PD-1 APC-Cyanine7 clone: 29 F.1A12 Catalog #: 1352224
BioLegend CD138 PE-Cyanine7 clone: 281-2 Catalog #: 142514
BioLegend CD23 PerCP-Cyanine5.5 clone: B3B4 Catalog #: 101618
BioLegend IA/IE APC-Cyanine7 clone: M5/114.15.2 Catalog #: 107628
BioLegend IA/IE APC clone: M5/114.15.2 Catalog #: 107614
BioLegend CXCR3 eFluor 450 clone: CXCR3-173 Catalog #: 126529
BioLegend CD4 PerCP-Cyanine5.5 clone: GK1.5 Catalog #: 100434
BioLegend CD8 BV605 clone: 53-6.7 Catalog #: 100743
BioLegend Ly6G BV605 clone: 1A8 Catalog #: 127612

BioLegend Ly6C FITC clone: HK1.4 Catalog #: 128006
BioLegend F4/80 PE clone: BM8 Catalog #: 123110
BioLegend CD11c APC-Cyanine7 clone: N418 Catalog #: 117352
BioLegend CD45 APC clone: 30-F11 Catalog #: 103112
Pblassay IFNa FITC clone: RMMA-1 Catalog #:22100-3

## Molecular cloning

Mouse cDNA of *Tlr7* (NM_001290755.1), *Tlr9* (NM_031178.2), *Abhd17a* (NM_145421.2), *Abhd17b* (XM_011247228.4), *Ppt1* (NM_008917.3), *Zdhhc1* (NM_175160.5), *Zdhhc2* (NM_178395.4), *Zdhhc3* (NM_026917.6), *Zdhhc4* (NM_028379.5), *Zdhhc5* (NM_144887.4), *Zdhhc6* (NM_025883.4), *Zdhhc7* (NM_133967.3), *Zdhhc8* (NM_172151.4), *Zdhhc9* (NM_172465.4), *Zdhhc11* (NM_027704.3), *Zdhhc12* (NM_025428.2), *Zdhhc13* (NM_028031.3), *Zdhhc14* (NM_146073.3), *Zdhhc15* (NM_175358.4), *Zdhhc16* (NM_023740.3), *Zdhhc17* (NM_172554.2), *Zdhhc18* (NM_001017968.2), *Zdhhc19* (NM_199309.2), *Zdhhc20* (NM_029492.5), *Zdhhc21* (NM_026647.4), *Zdhhc22* (NM_001377025.1), *Zdhhc23* (NM_001007460.2) and *Zdhhc24* (NM_027476.3) were reverse transcribed with HiScript II 1st Strand cDNA Synthesis Kit (Vazyme, R212-02) from the mRNA of spleen. Human cDNA of *TLR9* (NM_017442.4), *ZDHHC3* (NM_016598.3), *ZDHHC17* (NM_015336.4), *ZDHHC18* (NM_032283.3) and *ZDHHC20* (NM_153251.4) were reverse transcribed with HiScript II 1st Strand cDNA Synthesis Kit (Vazyme, Cat#R212-02) from the mRNA of PBMCs. Flag (DYKDDDDK) or hemagglutinin (YPYDVPDYA) was fused to the C-terminus of the indicated target genes, and then, all the genes were cloned into the lentivirus package vector pWPI (Addgene, 12254).

Point mutations of the genes encoding *Tlr9* (C258A, C265A), *Zdhhc3* (C157S), *Zdhhc17* (C467S), *Zdhhc18* (C214S), *Zdhhc20* (C156S) were established with Mut Express II Fast Mutagenesis Kit V2 (Vazyme, Cat# C214-02). Flag (DYKDDDDK) or hemagglutinin (YPYDVPDYA) was fused to the C-terminus of the indicated target genes, and then, all the genes were cloned into the lentivirus package vector pWPI (Addgene, 12254).

To generate RAW264.7 *Ppt1*[-/-], RAW264.7 *Tlr9*[-/-] and RAW264.7 *Zdhhc3*[-/-] cells, sgRNA-targeted lentiCRISPR v2 was transduced into RAW264.7 cells. Puromycin (2 μg/mL) was added to the cells 24 h post-transduction and maintained for 2 weeks. On Day 14, individual cells were seeded in a 96-well plate, and cells were monitored for growth before genomic DNA extraction and analysis. Gene disruption was confirmed by immunoblot analysis of target proteins or by functional analysis. The guide RNA sequences (*Ppt1*: GCTGGTGATCTGG-CATGGGA; *Tlr9*: GCTGAAGCCTCATGGCCTGG; *Zdhhc3*: GCCATGTGGTTTATCCGAGA) were designed and synthesized after codon optimization by the Tsingke Biotechnology Synthesis Service and subcloned into a lentiCRISPR v2 plasmid (Addgene, 52961).

## Cell lines

All cell lines used in this study tested negative for mycoplasma. The 293 T (ATCC, CRL-3216) and RAW264.7 macrophage cell lines (ATCC, TIB-71) were cultured in complete DMEM. Human monocytic THP-1 cells (ATCC, TIB-202) were grown in complete RPMI-1640. The cells were cultured at 37 °C in a humidified atmosphere with 5% CO$_2$. For the 293 T cells, transient transfection was performed using polyethylenimine (PEI) (Polysciences, 24765). For lentiviral transduction of RAW264.7 cells, 293 T cells were transfected with the lentiviral vectors pWPI, psPAX2 (Addgene, 12260) and pMD2.G (Addgene, 12259) using polyethylenimine (PEI) (Polysciences, 24765). Forty-eight hours after transfection, the virus-containing culture supernatant was used to infect target cells. GFP$^+$ target cells were sorted with a BD FACSAria III flow cytometer.

## Immunoprecipitation

IP protocol was previously described[93]. Protein lysates (1 mg per 1 ml) were suspended in 1 ml of lysis buffer containing a protease inhibitor cocktail and incubated with 20 μL anti-Flag matrix (Sigma, A2220) or anti-HA matrix (Roche, 11815016001) overnight at 4 °C with end-over-end rotation. Unbound proteins were removed by washing 4 times with 1 ml of lysis buffer and with spinning at 2500 rcf for 1 min each time. Precipitated proteins were denatured in 2× reducing SDS-PAGE sample buffer and boiled for 5 min at 95 °C.

## Cell fractionation by sucrose density-centrifugation

Cells in 3 confluent 15-cm dishes were washed in ice-cold PBS, scraped in 10 ml SHB and pelleted by centrifugation. Cells were resuspended in 2 ml SHB buffer (0.25 M sucrose, 1 mM EDTA, 10 mM HEPES, pH = 7.4) plus protease inhibitor cocktail with EDTA (Roche) and 1 mM PMSF and disrupted by 30 strokes in a steel dounce homogenizer. The disrupted cells were centrifuged for 12 min at 1400 g to remove nuclei. Supernatants were loaded onto continuous sucrose gradients (percentage iodixanol: 0, 10, 20, 30) and ultracentrifuged in an SW41 rotor at 181,299 x g for 1 h (Optima L-90K Ultracentrifuge, Beckman Coulter). Twenty-two fractions of 420 μl were collected from top to bottom, and 100 μl of each fraction were denatured in SDS buffer for western blot analysis.

## Western blotting

Protein samples were run on SDS-PAGE with a 4–12% gradient gel and then transferred. The membranes were blocked with 5% dried milk and incubated with primary antibodies overnight. After washing with PBS with Tween 20, secondary antibodies were incubated with the membranes for 1.5 h. Bands were visualized by chemical composition using ChemiDoc Touch (Bio-Rad). The relative gray areas of different bands were calculated by Image J (Version: 2.1.0/1.53c).

The following primary antibodies used for Western blotting were all used at a dilution of 1:1000 (unless otherwise indicated):
Sigma Anti-Flag (host: Mouse) Catalog #: F1840
Roche Anti-HA (host: Rat) Catalog #: 11867423001
Abcam Calnexin (host: Rabbit) Catalog #: ab213243
Abcam Rab7 (host: Rabbit) Catalog #: ab137029
Abcam Tubulin (host: Mouse) Catalog #: ab78078 (1:5000 dilution)
Novus PPT1 (host: Rabbit) Catalog #: NBP2-93840
Novus TLR9 (host: Mouse) Catalog #: NBP2-24729
CST anti-biotin, HRP-linked Catalog #: 7075 S
Abmart GM130 (host: Rabbit) Catalog #: T55142
CST EEA1 (host: Rabbit) Catalog #: 2411
Abcam goat anti-mouse IgG H&L (HRP) Catalog #: ab6789 (1:5000 dilution)
Abcam goat anti-rabbit IgG H&L (HRP) Catalog #: ab6721 (1:5000 dilution)
Abcam goat anti-rat IgG H&L (HRP) Catalog #: ab97057 (1:5000 dilution)

## ELISA

Supernatants were collected 16 h after TLRs stimulation. Samples were placed on precoated plates with anti-mouse TNFα (Thermo Fisher Scientific, 88-7324-88), anti-IFNα (Invitrogen, BMS6027), and anti-human IFNα (DAKEWE, 1110012) antibodies according to the manufacturer's instructions.

## RNA extraction and quantitative PCR

TRIzol reagent (Genstar, P118-05) and chloroform were added to homogenize single cells, and then, RNA precipitation, washing and resuspension were performed following the manufacturer's protocol. The extracted RNA was used for reverse transcription according to the manufacturer's protocol (HiScript III RT SuperMix for qPCR, Vazyme, R323-01). Quantitative RT-PCR analysis was performed with SYBR Select Master Mix (Genstar, A301-10) using StepOne Plus (Life Sciences). All data were normalized to *β-Actin* expression.

The following primers were used for the quantification of transcripts by real-time quantitative PCR:

*Ppt1*: forward, 5'-TTGTGGACCCTGTCGACTCT-3',
reverse, 5'-GATGGTCCCCTTCCTTAGCC-3';
*Ifna*: forward, 5'-AACCTCCTCTGACCCAGGAA-3',
reverse, 5'-GGCTCTCCAGACTTCTGCTC-3';
*β-actin*: forward, 5'-GTGACGTTGACATCCGTAAAGA-3',
reverse, 5'-GCCGGACTCATCGTACTCC-3'

## Immunological Genome Project

The RNA-seq datasets that support the findings of this study are publicly available from The Immunological Genome Project (ImmGen) (https://www.immgen.org), and these data were directly downloaded from the website, after searching for the appropriate gene name at http://rstats.immgen.org/Skyline/skyline.html.

## Statistical analysis

Statistical analysis was performed in Prism 8 (GraphPad) software. For murine samples, comparisons between two groups were performed with two-tailed Student's $t$ test. For human samples, statistical analysis was performed with a two-tailed Wilcoxon matched-pair signed rank test.

## Reporting summary

Further information on research design is available in the Nature Portfolio Reporting Summary linked to this article.

## Data availability

No novel datasets or reagents are generated in this study. All data supporting the findings of this study are included in the manuscript and its supplementary files are available. Source data are provided with this paper.

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

## Acknowledgements

We thank Dr. Juncao Cai for plasmids and Dr. Jun Cui for HSV-1. We are grateful to Dr. Vanja Sisirak and Dr. Xiaojun Xia for critically reading of the manuscript. We thank Dr. Weibin Cai for technical assistance at the animal facilities. This research was supported by the National Key R&D Program of China (2018YFA0508300, to C.Y.Y.), the Shanghai Laboratory Animal Program 23141902200 (to B.L.), the National Natural Science Foundation of China (32250610206 to C.Y.Y.), the Guangdong Innovative and Entrepreneurial Research Team Program (2016ZT06S638 to C.Y.Y.), the 111 Project (Grant No. B12003 to C.Y.Y.), the National Natural Science Foundation of China (82271819 and 82071820 to J.Z.), the National Natural Science Foundation of China (32170881 to B.L.), the Shanghai Pujiang Program (21PJ1414500 to B.L.), and the Shanghai Municipal Science and Technology Major Project (HS2021SHZX001 to B.L.).

## Author contributions

H.N. and K.Y. designed and performed the experiments, analyzed and interpreted the data, and wrote the manuscript. Y.W., L.W., J.H., Y.X. and H.C. performed the experiments. B.L. designed and supervised the TLR9 mechanism studies, interpreted the data, and wrote the manuscript. C.Y.Y. designed the mouse experiments, oversaw protocol logistics, interpreted the data, wrote the manuscript, contributed to funding, and supervised the entire study. J.Z. designed and supervised human experiments, oversaw protocol logistics, interpreted the data, wrote the manuscript, contributed to funding, and supervised the entire study.

## Competing interests

The authors declare no competing interests.
