## [Peer review file · Nature Communications]

REVIEWER COMMENTS

Reviewer #1 (expert in innate immunity and post-translational modifications):

The manuscript titled with "Thioesterase PPT1 regulates systemic lupus erythematosus via an endosomal TLR9 palmitoylation cycle" clarifies that thioesterase PPT1 plays a significant regulatory role in systemic lupus erythematosus via regulating the palmitoylation of TLR9. In addition, the PPT1 inhibitor HDSF is able to alleviate symptoms of SLE by interfering the palmitoylation modification cycle of TLR9. Admittedly, the authors made some efforts to reveal regulatory role of PPT1 for SLE, identified palmitoylation modifications of TLR9 and its major writer ZDHHC3 and eraser PPT1, and attempted to connect the above findings into a chain of regulatory mechanism that can significantly affect SLE pathogenesis. Unfortunately, it seems the manuscript lacks of sufficient solid data to support several important link of this chain. In particular, the specific physicochemical mechanism by which the palmitoylation modification of TLR9 affects its binding to ligands remain unclear. Although authors draw relatively cautious conclusion, the lack of exploration of underlying mechanism makes this paper probably hardly reach the standard of the journal. Therefore, acceptance of this paper is not recommended for the time being. The following are some issues of concern.

There are several concerns, below I listed several main and minor ones:

Major Concerns

1. What exactly is the molecular mechanism by which palmitoylation modification of TLR9 affects the function of TLR9 in SLE? According to the data in Fig 7, either decreasing or increasing the palmitoylation modification of TLR9 leads to the reduced binding of TLR9 to its ligand. This suggested that it is actually the disruption of palmitoylation modification cycle that leads to the reduction of ligand binding by TLR9. However, this is still only a phenomenon and the underlying mechanism has not been explored. Is vesicular transport of TLR9 between Golgi and lysosomes affected? More sufficient reliable data are needed for further exploration.
2. For Fig. S3A and Fig. 3B, the Ppt1 transcript level is higher in macrophages and pDCs of SLE and again upregulated after CpG A stimulation. Does these imply a disturbed cycle of palmitoylation modification of TLR9 which leading to a reduction of Ifna transcript level at 4-24 hrs and the associated attenuation of the inflammatory response? This seems to be a benign feedback regulation, why additional PPT1 inhibitors need to be imposed for interventional therapy? It is suggested to detect and compare the differences in PPT1 expression levels in macrophages and pDCs from healthy volunteers and SLE patients, as well as changes in response to CpG A stimulation, and to detect differences and changes in TLR9 palmitoylation modification and inflammatory factor secretion.
3. For Fig. 4E and Fig. 7B, there are weaker cleavage band of Mut2 (C258A+C265A) mutant of TLR9 than WT TLR9 in input. Can you give a reasonable explanation and corresponding data to prove it? Is it due to the loss of palmitoylation or something else? And for Fig. 6D, there are also weaker cleavage band of TLR9 in the group Ppt1 knockdown.

Minor Concerns

4. For all panels that blot TLR9 FL bands and cleavage bands on same membranes, it should be indicated which band was intercepted when performing grayscale statistics.
5. For Fig. 5 and Fig. 6, the titles are not rigorous as authors only based on previous studies stating that ZDHHC3 is mainly localized to Golgi and PPT1 is mainly localized to lysosome, thus speculating that the sites where their regulation of TLR9 palmitoylation occurs are Golgi and lysosome, but direct experimental data (e.g. PLA proximity linkage reaction or immunofluorescence staining containing organelle markers) are not provided for proof.
6. For Fig. 7 (7A, 7C, 7E, 7G), it is recommended to detect the difference and changes of TLR9 palmitoylation levels to provide a more comprehensive and compact logical correlation of the data.

Reviewer #2 (expert in SLE):

This study provides extensive data, mostly in the murine system but also including PBMC and pDC data from patients with SLE, supporting regulation of the function of TLR9 by a palmitoylation cycle and suggests that mediators of that cycle might serve as therapeutic targets to inhibit type I interferon production in SLE.

The studies appear to be appropriately performed and supportive of the palmitoylation-related mechanisms as significant contributors to the capacity of TLR9 to respond to CpG ligands with induction of cytokines in pDCs and also in macrophages.

As the authors establish the rationale for their study with a discussion of the relative roles of TLR7 and TLR9 in the pathogenesis of SLE and the apparent distinct mechanisms of those two endosomal TLRs, with TLR7 responsive to RNA ligands and TLR9 responsive to DNA ligands, it is appropriate to relate their presented data to a consideration of how their study elucidates those distinct roles in disease. As noted by the authors, TLR7 gain of function can support development of SLE. A number of studies have indicated that autoantibodies that target RNA-binding proteins are associated with activation of the type I interferon pathway (suggesting a role for TLR7), and their immune complexes can induce production of interferon by pDCs through TLR7. In addition, TLR7 has been shown to be an important signaling pathway for differentiation of autoantibody producing B cells in SLE. As noted by the authors, TLR9 activation, at least in some murine studies, can provide protection from SLE. Given all of that, there remain important questions regarding the relative roles of TLR7 and TLR9 in SLE pathogenesis, including the issue of expression of which TLR in which cells, e.g., pDCs or B cells, provides the most important contributions to disease (IFN production and/or autoantibody production) and which ligands, RNA or DNA, are most involved in induction of interferon and/or B cell differentiation.

The authors' data support TLR9 function on pDCs as important for interferon-alpha production but do not directly study the impact of TLR9 function on B cell differentiation. While anti-dsDNA antibodies are reduced and their kidney deposition is reduced in PPT1-deficient mice, total IgG is also significantly reduced, so it is not clear if there is a specific impact on DNA-targeted specificities. While the authors show that a TLR7 ligand does not alter TLR9 palmitoylation, they do not measure the impact on TLR7 palmitoylation or address whether the palmitoylation cycle affects TLR7. The 2-BP experiments perhaps suggest that the TLR9 palmitoylation cycle may be more relevant to pDC functions than to B cell functions, but the authors do not pursue that point. They do not mention TLR7 in their Discussion.

Given the fact that the authors set up their study in the context of the roles of TLR7 and TLR9 in SLE pathogenesis, it would be very helpful for the authors to:

- address whether TLR7 is also involved in a similar palmitoylation cycle
- present data on whether PPT1 inhibition/deficiency reduces anti-RNA-binding protein autoantibodies (e.g., anti-Sm or -RNP) to the same extent as anti-dsDNA
- address more specifically the impact of the palmitoylation cycle on B cells activated through TLR9.
- Discuss whether their data provide new insights into the relative roles of TLR7 and TLR9 in SLE pathogenesis, particularly with regard to the proposed protective role of TLR9.

Reviewer #3 (expert in TLRs in lupus):

Hai Ni et al here present interesting and fairly thorough investigations of S-palmitoylation of TLR9 and the effect of a palmitoylation / depalmitoylation cycle on TLR9 signaling. They further show using knockouts and chemical inhibitors that palmitoylation, more generally, is important for SLE pathogenesis in a mouse model. Overall, these studies open up an interesting new direction for investigating how TLRs are regulated both generally and particularly in SLE.

Major comments:

1) I think the title of the work somewhat overstates the conclusions that can be drawn from this data. The authors show convincingly that TLR9 is palmitoylated. They also show convincingly that knocking out the depalmitoylation enzyme PPT1 in the B6.Sle1.yaa model has an effect on immune activation in the lupus model, as does chemical interference with this pathway. However, a great many proteins are palmitoylated, and so it cannot be concluded that the palmitoylation / depalmitoylation of TLR9 itself is solely responsible for the other effects seen with these inhibitors and genetic manipulations of the palmitoylation pathway enzymes, as the title implies. For example, Myd88 is palmitoylated, as the authors note; this could affect signaling via TLR7, which is particularly important in the yaa model that includes a TLR7 gene duplication onto the y-chromosome. Is TLR7 signaling (for example, IFN α production by pDC in response to TLR7 ligand) different in PPT1 knockout cells or HDSF-treated cells / mice? Are anti-RNA autoantibodies also affected in either of these models?

2) The authors indicate that mutation of the two cysteine sites to alanine reduced (but did not completely eliminate) palmitoylation of TLR9 protein and affected signaling by TLR9. However, these mutations also affected the amount of cleaved versus full-length TLR9 in the input. Is palmitoylation affecting the trafficking / localization and proteolytic processing of TLR9, or is palmitoylation more directly affecting signaling? Perhaps some quantification of cleaved-to-full-length ratio could be provided for the input, as well as for the CpG-bound fraction, in 7B (and/or 4E). It looks like cleavage is efficient in both the Zddhc3 and Ppt1 knockouts (Fig 7D, F inputs) but it is hard to be sure from the blots.

3) I think the BDCA2-DTR mixed chimera experiment is not explained clearly. It could help if in supplemental figure 3C the top donor was labelled as "CD45.1+ WT or CD45.1+ Ppt1 $^{-/-}$ " which I think must be the design here. Under the conditions of this experiment, after DT depleting the BDCA2-DTR Ppt $^{+/+}$ cells, still half of all other hematopoietic cells will be Ppt $^{-/-}$ from the CD45.1 donor. It is not, as stated in the text, that "only pDCs possessed PPT1 deficiency," rather, you are depleting the WT pDC from the BDCA2-DTR donor, leaving only PPT1-deficient pDC but also half of everything else PPT-deficient. While it is likely that the pDC are the source of the difference in serum IFN α , I don't think that can actually be concluded from this experiment. To control for this, you could have included controls such as 100% BDCA2-DTR PPT1 $^{+/+}$ and 100% BDCA2-DTR PPT1 $^{-/-}$; when treated with DT, pDC would be eliminated completely and the remaining hematopoietic cells would be 100% WT or 100% PPT1 $^{-/-}$. If there were any differences between those two groups, it could point to a non-pDC source of PPT1-dependent IFN α .

Minor point:

1) I wonder if the labeling of murine DHHCs screened in Figure 5A is quite correct? I think there is not a mammalian Zddhc10 gene (and indeed in this paper's methods describing cloning of the mouse DHHCs this one is skipped) while Zddhc23 seems to be omitted from the figure but is described in the methods.

We would like to thank all reviewers for their constructive criticism of our manuscript.
We have addressed your concerns in the revised manuscript with additional
experiments. The changes are described below and also underlined in the revision.

**Reviewer #1 (expert in innate immunity and post-translational modifications):**

*The manuscript titled with “Thioesterase PPT1 regulates systemic lupus*
*erythematosus via an endosomal TLR9 palmitoylation cycle” clarifies that thioesterase*
*PPT1 plays a significant regulatory role in systemic lupus erythematosus via*
*regulating the palmitoylation of TLR9. In addition, the PPT1 inhibitor HDSF is able to*
*alleviate symptoms of SLE by interfering the palmitoylation modification cycle of TLR9.*
*Admittedly, the authors made some efforts to reveal regulatory role of PPT1 for SLE,*
*identified palmitoylation modifications of TLR9 and its major writer ZDHHC3 and eraser*
*PPT1, and attempted to connect the above findings into a chain of regulatory*
*mechanism that can significantly affect SLE pathogenesis. Unfortunately, it seems the*
*manuscript lacks of sufficient solid data to support several important link of this chain.*
*In particular, the specific physicochemical mechanism by which the palmitoylation*
*modification of TLR9 affects its binding to ligands remain unclear. Although authors*
*draw relatively cautious conclusion, the lack of exploration of underlying mechanism*
*makes this paper probably hardly reach the standard of the journal. Therefore,*
*acceptance of this paper is not recommended for the time being. The following are*
*some issues of concern.*

*There are several concerns, below I listed several main and minor ones:*

*Major Concerns*

*1. What exactly is the molecular mechanism by which palmitoylation modification of*
*TLR9 affects the function of TLR9 in SLE? According to the data in Fig 7, either*
*decreasing or increasing the palmitoylation modification of TLR9 leads to the reduced*
*binding of TLR9 to its ligand. This suggested that it is actually the disruption of*
*palmitoylation modification cycle that leads to the reduction of ligand binding by TLR9.*
*However, this is still only a phenomenon and the underlying mechanism has not been*
*explored. Is vesicular transport of TLR9 between Golgi and lysosomes affected? More*
*sufficient reliable data are needed for further exploration.*

Thank you for this key insight about TLR9 vesicular transport. We were also puzzled
that the disruption of palmitoylation or depalmitoylation yielded similar phenotype.
Therefore, we looked harder at TLR9 trafficking, as you pointed out.

Palmitoylation in the Golgi:

After performing cellular fractionation experiments, we found that DHHC3-deficient

cells retained more full-length TLR9 in the Golgi fractions, indicated by GM130
 enrichment (**Fig. 7I**, and shown below). Thus, palmitoylation is required for the
 trafficking of TLR9 from the Golgi to early endosomes (indicated by EEA1). Lower
 trafficking efficiency of full length of mTLR9 might result in decrease of TNF secretion
 that was observed in DHH3-deficient cells (**Fig. 7C**).

Depalmitoylation in endosomes:

In endosomes, the release of cleaved TLR9 from UNC93B1 is required for receptor
 signaling, whereas TLR7 signaling do not require such release (Majer O., et al, *Nature*,
 2019). Using Co-IP experiment to probe the amount of retained TLR9 on UNC93B1,
 we found that there was more TLR9 bound to UNC93B1 in HDSF-treated cells (**Fig.**
 **7J**, and shown below)

However, there was subtle difference in TLR9-UNC93B1 binding affinity when we
 compared WT with PPT1-deficient cells (data not shown), suggesting that other
 depalmitoylation enzymes may compensate PPT1 deficiency in this process.
 Therefore, we think that HDSF may inhibit other palmitoyl thioesterases and have a
 broader target besides PPT1. Previous literature had shown that
 Hexadecylfluorophosphonate (HDFP), which has a similar structure with HDSF, could
 inhibit many serine hydrolases including PPT1 (see below).

Modified from *Martin BR, Wang C, Adibekian A, Tully SE, Cravatt BF. Global profiling of*
 *dynamic protein palmitoylation. Nat Methods. 2011 Nov 6;9(1):84-9. doi:*
 *10.1038/nmeth.1769. PMID: 22056678; PMCID: PMC3248616.*

Thus far, only 5 lysosomal enzymes were known to depalmitoylate proteins. We
 screened PPT1, ABHD17a, and ABHD17b. It is also possible that other unidentified
 thioesterases could also depalmitoylate TLR9. Accordingly, we have softened our
 previous conclusion that PPT1 was only depalmitoylation enzyme in the results (**p.14**)
 and discussed this result in the discussion (**p.18**).

In summary, TLR9 palmitoylation regulates TLR9 trafficking from the Golgi to
 endosomes, while TLR9 depalmitoylation controls the release of TLR9 from UNC93B1
 in endosomes. Inhibition of either step would result in less free cleaved TLR9 (the
 functional form) in endosomes and therefore reduced signaling.

*2.For Fig. S3A and Fig. 3B, the Ppt1 transcript level is higher in macrophages and*
 *pDCs of SLE and again upregulated after CpG A stimulation. Does these imply a*
 *disturbed cycle of palmitoylation modification of TLR9 which leading to a reduction of*
 *Ifna transcript level at 4-24 hrs and the associated attenuation of the inflammatory*
 *response? This seems to be a benign feedback regulation, why additional PPT1*
 *inhibitors need to be imposed for interventional therapy?*

This moderate (~80%) upregulation was indeed unexplored. Regarding your
 hypothesis about the feedback mechanism, we checked IFN α production by pDCs at
 4 hours and 24 hours, and found that there is no detectable IFN α secretion at earlier
 timepoint (**Fig. S3B**, and shown below). IFN α is probably the earliest cytokine
 produced by pDCs. Thus, the upregulation of *Ppt1* transcripts at 4 hours is unlikely to
 be a feedback mechanism in response to IFN α .

It is also very interesting to investigate the importance of this upregulation in terms of TLR9 palmitoylation. Our ABE assays were always performed at 4 hours after stimulation. Thus, we performed ABE assays to compare TLR9 palmitoylation state in *Ppt1*^{+/+} or *Ppt1*^{-/-} Flt3L-pDCs at late timepoints (24 hours) after CpG A stimulation. We could see that TLR9 palmitoylation was still enhanced in *Ppt1*^{-/-} pDCs (**Fig. 6E**, and shown below), albeit to a lesser degree than 4 hours. This suggested that PPT1 expression is positively associated with TLR9 depalmitoylation.

Respectfully, we think that this is a not a feedback mechanism, but a programmed mechanism to maximize IFN-I production in pDCs. As you pointed out, increased Ppt1 transcripts was associated with higher IFN α transcripts (**Fig. 3B**). Consistently, PPT1 deficiency or inhibitors suppressed IFN α (**Fig. 3C, S3D, 3F and S3I**). Collectively, these experiments suggested that PPT1 expression is *positively* associated with IFN-I production. After TLR9 stimulation, pDCs may need to express more PPT1 to depalmitoylate TLR9 and facilitate a stronger IFN α response. We have changed our results section about PPT1 upregulation with the new results here (**p.9**).

Regarding the rationale for PPT1 inhibitors as therapy, we demonstrated that PPT1 deficiency suppressed immune response in autoimmunity and viral infections (**Fig.1A-H and Fig. 3D**). These *in vivo* results support the theory that PPT1 expression

facilitates TLR9 response and autoimmunity. Therefore, PPT1 inhibitors should work
 and indeed worked as interventional therapy (**Fig. 2B-K**).

*It is suggested to detect and compare the differences in PPT1 expression levels in*
 *macrophages and pDCs from healthy volunteers and SLE patients, as well as changes*
 *in response to CpG A stimulation, and to detect differences and changes in TLR9*
 *palmitoylation modification and inflammatory factor secretion.*

This is a really good suggestion. We tried our best to compare PPT1 expression in
 macrophages and pDCs from humans. However, the number of pDCs in patient
 PMBCs is simply too low (<500 cells) to warrant any trustworthy experiments. In our
 clinical protocol, we are only allowed to obtain 20ml per patient, because they are also
 needed to provide lots of blood for other regular lab tests. We do appreciate this idea
 and included this suggestion in our revised discussion (**p.21**).

*3.For Fig. 4E and Fig. 7B, there are weaker cleavage band of Mut2 (C258A+C265A)*
 *mutant of TLR9 than WT TLR9 in input. Can you give a reasonable explanation and*
 *corresponding data to prove it? Is it due to the loss of palmitoylation or something else?*

Yes, that is a sharp observation. There were no such weakened cleavage bands in our
 DHHC3 (**Fig. 5C, 7D, S7A**, input panels) or PPT1-deficient cells (**Fig. 6F, 7F, S7A**,
 input panels). However, there is weaker cleavage bands in TLR9-Mut2 cells in the input
 panel, before CpG-biotin pulldown (**Fig. S7A** and shown below).

We also performed cellular fractionation experiments on Mut2 cells, mTLR9^{Mut2} had no
 such trafficking defect, suggesting that its mechanism may not be entirely the same as
 in palmitoylation-defect cells (**Fig. S7B**, compared to **Fig. 7I**, and shown below).

This result indicated that the TLR9-Mut2 phenotypes are not entirely due to the loss of
 palmitoylation. One possible explanation could be that these two mutations play a role
 in TLR9 cleavage process. Multiple lysosomal proteases are required for TLR9
 cleavage. These two mutations may result in an altered structure, or may interfere with
 this enzymatic process. Given that TLR9 is a highly conserved protein, these two sites
 may have additional function beyond palmitoylation. We have included this
 discrepancy in the results (p.15) and discussion (p.17).

And for Fig. 6D, there are also weaker cleavage band of TLR9 in the group *Ppt1*
 knockdown.

Good eye! We reanalyzed all 3 previous blots (run 1-3, shown below), and found there
 is actually no significant difference in cleaved bands in PPT1-deficient cells. To make
 sure, we performed 2 additional runs (run 3-4) and still found no difference (shown
 below). This could be an inadvertent bad choice of non-representative blots. We
 apologize for this mistake and will replace it with run 2, a more representative blot for
 all 5 repeats (now in Fig.6F). A more complete quantification of cleaved TLR9 in input
 panel is provided in Fig. S7A.

*Minor Concerns*

*4. For all panels that blot TLR9 FL bands and cleavage bands on same membranes, it*
*should be indicated which band was intercepted when performing grayscale statistics.*

Thank you for this wonderful suggestion. We have now included boxes in all panels
with TLR9, indicating which bands were intercepted.

*5. For Fig. 5 and Fig. 6, the titles are not rigorous as authors only based on previous*
*studies stating that ZDHHC3 is mainly localized to Golgi and PPT1 is mainly localized*
*to lysosome, thus speculating that the sites where their regulation of TLR9*
*palmitoylation occurs are Golgi and lysosome, but direct experimental data (e.g. PLA*
*proximity linkage reaction or immunofluorescence staining containing organelle*
*markers) are not provided for proof.*

We apologize for the misplaced citations that confused you. We only cited the reviews
stating the general conclusions but have no direct experimental data. We have not
performed these experiments because many previous publications had directly shown
their locations.

Regarding DHHC3:

1) Keller CA, Yuan X, Panzanelli P, et al. *The gamma2 subunit of GABA(A) receptors is a*
*substrate for palmitoylation by GODZ. J Neurosci. 2004;24(26):5881-5891.*
*doi:10.1523/JNEUROSCI.1037-04.2004*

Legend: (C-E), Primary cultured cortical neuron double labeled for primary Anti-DHHHC3
(C, red) and Golgi 58 kDa (D, green). Colocalization is shown in yellow (E).

2) Lu D, Sun HQ, Wang H, et al. *Phosphatidylinositol 4-kinase IIa is palmitoylated by Golgi-*
*localized palmitoyltransferases in cholesterol-dependent manner. J Biol Chem.*
*2012;287(26):21856-21865. doi:10.1074/jbc.M112.348094*

Legend: Colocalization of endogenous DHHCs with EGFP-PI4KII α in Golgi. HeLa cells
 were transfected with EGFP-PI4KII α and then labeled with anti-TGN46 (Golgi marker) and
 anti-DHHC3 antibodies to detect endogenous proteins.

3) Kilpatrick CL, Murakami S, Feng M, et al. Dissociation of Golgi-associated DHHC-type
 Zinc Finger Protein (GODZ) and Sertoli Cell Gene with a Zinc Finger Domain- β (SERZ- β)-
 mediated Palmitoylation by Loss of Function Analyses in Knock-out Mice. *J Biol Chem.*
 2016;291(53):27371-27386. doi:10.1074/jbc.M116.732768

Legend: HEK293T cells were transfected with GFP-N3 (cis-Golgi apparatus marker) and
 then labeled with endogenous anti-GODZ (also known as DHHC3). Note that endogenous
 GODZ were colocalized with the cis-Golgi marker GFP-N3.

4) Niki Y, Adachi N, Fukata M, et al. S-Palmitoylation of Tyrosinase at Cysteine⁵⁰⁰
 Regulates Melanogenesis. *J Invest Dermatol.* 2023;143(2):317-327.e6.
 doi:10.1016/j.jid.2022.08.040

Figure 5. Intercellular localization of DHHC3 in normal human epidermal melanocyte
 (NHEMs). NHEMs were transfected with plasmid encoding Myc-tagged DHHC3. Two days
 after transfection, they were fixed, permeabilized, and stained with GM130 antibody (cis-

Golgi apparatus marker) and a Myc antibody. Bar = 20 mm.

Regarding PPT1:

1) Verkruyse LA, Hofmann SL. Lysosomal targeting of palmitoyl-protein thioesterase.

*J Biol Chem.* 1996;271(26):15831-15836. doi:10.1074/jbc.271.26.15831

FIG. 4. Percoll density gradient analysis of subcellular fractions of Madin-Darby bovine

kidney cells. Upper panel, marker enzyme analysis. The percent of the total activity is

plotted as a function of distance from the bottom of the gradient (fraction 1). Lower

panel, immunoblot of Percoll density gradient fraction using anti-bovine PPT antibodies.

2) Bagh MB, Peng S, Chandra G, et al. Misrouting of v-ATPase subunit V0a1

dysregulates lysosomal acidification in a neurodegenerative lysosomal storage

disease model. *Nat Commun.* 2017;8:14612. Published 2017 Mar 7.

doi:10.1038/ncomms14612

Figure 4 | Endosomal and lysosomal localization of Ppt1. (a) Colocalization of Ppt1
 with lysosome marker (LAMP 1) in WT and *Cln1*^{-/-} brain cells (n 1/4 34). (b)
 Colocalization of Ppt1 with early endosome marker (EEA1) in WT and *Cln1*^{-/-} cells (n
 1/4 6). (c) Colocalization of Ppt1 with late endosome marker (Rab 9) in WT and *Cln1*^{-/-}
 cells (n 1/4 5). Colocalization of Ppt1 with the endosomal markers in WT cells was
 assessed using the Manders' colocalization coefficients M1 and M2. Scale bars, 5 mm.

In our revised manuscript, we have now cited primary literature that directly
 demonstrated their localizations (**p.12** and **p.13**).

*6.For Fig. 7 (7A, 7C, 7E, 7G), it is recommended to detect the difference and changes*
 *of TLR9 palmitoylation levels to provide a more comprehensive and compact logical*
 *correlation of the data.*

Thank you for the suggestion to correlate ligand concentration with TLR9
 palmitoylation. These figures are titration curves and contained 45 different test
 conditions. Before conducting 45 different ABE assays, we are concerned that ABE
 assays, by principle, are oversaturated, and may not be sensitive enough to detect
 difference in palmitoylation when ligands concentration is diluted serially. We tested
 two concentrations of CpG B, and were unable to see much difference in TLR9
 palmitoylation levels (shown below). Therefore, we think that ABE assay is unable to
 accurately reflect the amount of TLR9 palmitoylation in a ligand concentration-
 dependent manner.

Nevertheless, we agree that the data should be more compact, and only included one
 ligand concentration, instead of the entire titration curve to streamline the figures (**Fig.**
 **7A, 7C, 7E**)

*Reviewer #2 (expert in SLE):*

*This study provides extensive data, mostly in the murine system but also including*
*PBMC and pDC data from patients with SLE, supporting regulation of the function of*
*TLR9 by a palmitoylation cycle and suggests that mediators of that cycle might serve*
*as therapeutic targets to inhibit type I interferon production in SLE.*

*The studies appear to be appropriately performed and supportive of the palmitoylation-*
*related mechanisms as significant contributors to the capacity of TLR9 to respond to*
*CpG ligands with induction of cytokines in pDCs and also in macrophages.*

*As the authors establish the rationale for their study with a discussion of the relative*
*roles of TLR7 and TLR9 in the pathogenesis of SLE and the apparent distinct*
*mechanisms of those two endosomal TLRs, with TLR7 responsive to RNA ligands and*
*TLR9 responsive to DNA ligands, it is appropriate to relate their presented data to a*
*consideration of how their study elucidates those distinct roles in disease. As noted by*
*the authors, TLR7 gain of function can support development of SLE. A number of*
*studies have indicated that autoantibodies that target RNA-binding proteins are*
*associated with activation of the type I interferon pathway (suggesting a role for TLR7),*
*and their immune complexes can induce production of interferon by pDCs through*
*TLR7. In addition, TLR7 has been shown to be an important signaling pathway for*
*differentiation of autoantibody producing B cells in SLE. As noted by the authors, TLR9*
*activation, at least in some murine studies, can provide protection from SLE. Given all*
*of that, there remain important questions regarding the relative roles of TLR7 and TLR9*
*in SLE pathogenesis, including the issue of expression of which TLR in which cells,*
*e.g., pDCs or B cells, provides the most important contributions to disease (IFN*
*production and/or autoantibody production) and which ligands, RNA or DNA, are most*
*involved in induction of interferon and/or B cell differentiation.*

*The authors' data support TLR9 function on pDCs as important for interferon-alpha*
*production but do not directly study the impact of TLR9 function on B cell differentiation.*
*While anti-dsDNA antibodies are reduced and their kidney deposition is reduced in*
*PPT1-deficient mice, total IgG is also significantly reduced, so it is not clear if there is*
*a specific impact on DNA-targeted specificities. While the authors show that a TLR7*
*ligand does not alter TLR9 palmitoylation, they do not measure the impact on TLR7*
*palmitoylation or address whether the palmitoylation cycle affects TLR7. The 2-BP*
*experiments perhaps suggest that the TLR9 palmitoylation cycle may be more relevant*
*to pDC functions than to B cell functions, but the authors do not pursue that point. They*
*do not mention TLR7 in their Discussion.*

*Given the fact that the authors set up their study in the context of the roles of TLR7*
*and TLR9 in SLE pathogenesis, it would be very helpful for the authors to:*

-address whether TLR7 is also involved in a similar palmitoylation cycle

Thank you for raising the importance of TLR7. We agree that TLR7 palmitoylation cycle
should be examined more closely, especially because TLR7 does not have the same
release mechanism as TLR9. In short, we found that TLR7 is not in a similar
palmitoylation cycle as TLR9.

TLR7 is palmitoylated:

Using click chemistry, we confirmed that TLR7 was palmitoylated (**Fig. 4F** and shown
below).

ABE assays showed the same conclusion (**Fig. 4G** and shown below).

TLR7 is not palmitoylated by DHHC3

DHHC3 overexpression could not increase TLR7 palmitoylation. In comparison,
DHHC3 could increase TLR9 palmitoylation signal by 8-fold (**Fig. 5A**). Other DHHCs
might be responsible for TLR7 palmitoylation (**Fig. 5D** and shown below).

TLR7 is never depalmitoylated:

ABE assays showed that TLR7 was not depalmitoylated after R848 activation (**Fig. 6B**
 and shown below). In contrast, TLR9 was quickly depalmitoylated after CpG activation
 and stayed depalmitoylated even after 24 hours (**Fig 6A** and **6E**).

PPT1 does not depalmitoylate TLR7

Overexpression of PPT1 do not cause TLR7 to depalmitoylate (**Fig. 6D** and shown
 below). In contrast, in the same experiments, PPT1 depalmitoylated TLR9 (**Fig. 6C**).

HDSF does not regulate TLR7 palmitoylation

Accordingly, PPT1 inhibitor HDSF did not change mTLR7 palmitoylation level (**Fig.**
 **S6H** and shown below). In contrast, HDSF strongly restored TLR9 palmitoylation (**Fig.**
 **6G**).

In conclusion, TLR7 is not involved in the same cycle of palmitoylation/depalmitoylation
 as TLR9. This may be partially due to the reason that cleaved TLR7 does not need to
 be released from UNC93B1, while TLR9 does. Combined with our new finding that
 depalmitoylation regulates the release of TLR9 from UNC93B1 (**Fig. 7J**). It makes
 sense that TLR7 is not involved in the depalmitoylation part of the cycle.

Given our main phenotype is revolved around PPT1, a depalmitoylation enzyme, it is
 less likely that the phenotype we observed in the PPT1-deficient mice is due to
 perturbed depalmitoylation of TLR7. However, the role of TLR7 should not be
 overlooked. We changed the main title, included above mentioned TLR7 data in the

results, changed the titles of **Figure 1** and **Figure 4**, and added appropriate discussion
(p.20).

-present data on whether PPT1 inhibition/deficiency reduces anti-RNA-binding protein
autoantibodies (e.g., anti-Sm or -RNP) to the same extent as anti-dsDNA

We tested the anti-RNP/Sm antibodies in serum from PPT1-deficient and HDSF-
treated mice. We found it was also downregulated (**Fig. 1C, 2D**, and shown below).

Therefore, TLR7 may play a role, albeit weaker, than TLR9 in the phenotype exhibited
in the PPT1-deficient mice. However, it is difficult to differentiate whether anti-RNP/Sm
autoantibodies are a direct consequence of blunted TLR7 response, or an indirect
effect as a result of reduced inflammation with an inhibited TLR9 response. We
changed the main title, included additional TLR7 data in results, changed the title of
**Figure 1** and **Figure 4**, and added appropriate discussion (p.20).

-address more specifically the impact of the palmitoylation cycle on B cells activated
through TLR9.

Great suggestion! We definitely overlooked the impact of the palmitoylation cycle on B
cells, because PPT1 expression in B cells was lower than pDCs and macrophages
(**Fig. S3A**).

The role of depalmitoylation in B cells:

We enriched B cells from Ppt1^{+/+} or Ppt1^{-/-} mice and examined their IL-6 production
with ELISA. We found that PPT1 had an impact on IL-6 production by B cells, though
the difference was very modest (**Fig. 3E**, and shown below).

Next, we examined Ppt1^{+/+} or Ppt1^{-/-} B cell proliferation after TLR9 stimulation. We also
found that there was a slight difference (**Fig. S3H**, and shown below).

PPT1 inhibitor HDSF had a stronger effect on B cell cytokine secretion and proliferation (Fig. 3F, S3J, and shown below). This pattern is consistent as in pDCs and macrophages (Fig. 3F).

The role of palmitoylation in B cells:

Unfortunately, we don't have DHHC3 knockout mice. Thus, we used 2-BP, a general palmitoylation inhibitor, on B cells. We found that 2-BP decreased IL-6 production in B cells (Fig. S7D, and shown below).

We also examined the effect of 2-BP in B cell proliferation, and there was a slight difference (Fig. S7E, and shown below).

In conclusion, both TLR9 palmitoylation cycles had an impact in B cell function, and B

cells may contribute partially to the phenotype we observed in the PPT1-deficient mice.
Thus, we have included these new data in the revised manuscript, changed the title of
Figure 3 (p.8) and discussion accordingly (p.21).

*-Discuss whether their data provide new insights into the relative roles of TLR7 and*
*TLR9 in SLE pathogenesis, particularly with regard to the proposed protective role of*
*TLR9.*

Thank you for the advice. We have now included our new data regarding TLR7 and B
cells in our discussion (p.20-21).

**Reviewer #3 (expert in TLRs in lupus):**

*Hai Ni et al here present interesting and fairly thorough investigations of S-*
*palmitoylation of TLR9 and the effect of a palmitoylation / depalmitoylation cycle on*
*TLR9 signaling. They further show using knockouts and chemical inhibitors that*
*palmitoylation, more generally, is important for SLE pathogenesis in a mouse model.*
*Overall, these studies open up an interesting new direction for investigating how TLRs*
*are regulated both generally and particularly in SLE.*

*Major comments:*

*1) I think the title of the work somewhat overstates the conclusions that can be drawn*
*from this data. The authors show convincingly that TLR9 is palmitoylated. They also*
*show convincingly that knocking out the depalmitoylation enzyme PPT1 in the*
*B6.Sle1.yaa model has an effect on immune activation in the lupus model, as does*
*chemical interference with this pathway. However, a great many proteins are*
*palmitoylated, and so it cannot be concluded that the palmitoylation / depalmitoylation*
*of TLR9 itself is solely responsible for the other effects seen with these inhibitors and*
*genetic manipulations of the palmitoylation pathway enzymes, as the title implies. For*
*example, Myd88 is palmitoylated, as the authors note;*

Yes, you are absolutely right that PPT1 may depalmitoylate many proteins. Our original
title states that PPT1 regulates SLE via TLR9, which may be too narrow. We have now
deleted “TLR9” from the title. The new title is “Thioesterase PPT1 regulates systemic
lupus erythematosus via an endosomal palmitoylation cycle”.

*this could affect signaling via TLR7, which is particularly important in the yaa model*
*that includes a TLR7 gene duplication onto the y-chromosome.*

Thank you for your suggestion. We explored more about the palmitoylation cycle of

TLR7. In short, TLR7 is not the same as TLR9, and PPT1 did not depalmitoylate TLR7.

TLR7 is palmitoylated:

Using click chemistry, we confirmed that TLR7 was palmitoylated (**Fig. 4F** and shown

below).

ABE assays showed the same conclusion (**Fig. 4G** and shown below).

TLR7 is not palmitoylated by DHHC3

DHHC3 overexpression could not increase TLR7 palmitoylation. In comparison,

DHHC3 could increase TLR9 palmitoylation signal by 8-fold (**Fig. 5A**). Other DHHCs

might be responsible for TLR7 palmitoylation (**Fig. 5D** and shown below).

TLR7 is never depalmitoylated:

ABE assays showed that TLR7 was not depalmitoylated after R848 activation (**Fig. 6B**
 and shown below). In contrast, TLR9 was quickly depalmitoylated after CpG activation
 and stayed depalmitoylated even after 24 hours (**Fig 6A** and **6E**).

PPT1 does not depalmitoylate TLR7

Overexpression of PPT1 do not cause TLR7 to depalmitoylate (**Fig. 6D** and shown
 below). In contrast, in the same experiments, PPT1 depalmitoylated TLR9 (**Fig. 6C**).

HDSF does not regulate TLR7 palmitoylation

Accordingly, PPT1 inhibitor HDSF did not change mTLR7 palmitoylation level (**Fig.**
 **S6H** and shown below). In contrast, HDSF strongly restored TLR9 palmitoylation (**Fig.**
 **6G**).

In conclusion, TLR7 is not involved in the same cycle of palmitoylation/depalmitoylation
 as TLR9. This may be partially due to the reason that cleaved TLR7 does not need to
 be released from UNC93B1, while TLR9 does. Combined with our new finding that
 depalmitoylation regulates the release of TLR9 from UNC93B1 (**Fig. 7J**). It makes
 sense that TLR7 is not involved in the depalmitoylation part of the cycle.

Given our main phenotype is revolved around PPT1, a depalmitoylation enzyme, it is
 less likely that the phenotype we observed in the PPT1-deficient mice is due to
 perturbed depalmitoylation of TLR7. However, the role of TLR7 should not be
 overlooked. We changed the title, included above mentioned TLR7 data in the results,

changed the title of **Figure 1** and **Figure 4**, and added appropriate discussion (p.20).

*Is TLR7 signaling (for example, IFN α production by pDC in response to TLR7 ligand)*
*different in PPT1 knockout cells or HDSF-treated cells / mice?*

We agree that the role of TLR7 was not explored. We measured IFN α secretion in
TLR7-activated PPT1-deficient pDCs. We found that TLR7 signaling is slightly affected
in PPT1-deficient pDCs (**Fig. 3C** and shown below).

*Are anti-RNA autoantibodies also affected in either of these models?*

We tested the anti-RNP antibodies in serum from PPT1-deficient and HDSF-treated
mice. We found it was also downregulated (**Fig. 1C, 2D**, and shown below).

Therefore, TLR7 may play a role, albeit weaker, than TLR9 in the phenotype exhibited
in the PPT1-deficient mice. However, it is difficult to differentiate whether anti-RNP/Sm
autoantibodies are a direct consequence of blunted TLR7 response, or an indirect
effect as a result of reduced inflammation with an inhibited TLR9 response. We
changed the main title, included additional TLR7 data in results, changed the title of
**Figure 1** and **Figure 4**, and added appropriate discussion (p.20).

*2) The authors indicate that mutation of the two cysteine sites to alanine reduced (but*
*did not completely eliminate) palmitoylation of TLR9 protein and affected signaling by*
*TLR9. However, these mutations also affected the amount of cleaved versus full-length*
*TLR9 in the input. Is palmitoylation affecting the trafficking / localization and proteolytic*
*processing of TLR9, or is palmitoylation more directly affecting signaling?*

Thank you for this key insight about TLR9 trafficking. We looked harder at TLR9
 trafficking, as you pointed out.

Palmitoylation in the Golgi:

After performing cellular fractionation experiments, we found that DHHC3-deficient
 cells retained more full-length TLR9 in the Golgi fractions, indicated by GM130
 enrichment (**Fig. 7I**, and shown below). Thus, palmitoylation is required for the
 trafficking of TLR9 from the Golgi to early endosomes (indicated by EEA1). Lower
 trafficking efficiency of full length of mTLR9 might result in decrease of TNF secretion
 that was observed in DHHC3-deficient cells (**Fig. 7C**).

Depalmitoylation in endosomes:

In endosomes, the release of cleaved TLR9 from UNC93B1 is required for receptor
 signaling, whereas TLR7 signaling do not require such release (Majer O., et al, *Nature*,
 2019). Using Co-IP experiment to probe the amount of retained TLR9 on UNC93B1,
 we found that there was more TLR9 bound to UNC93B1 in HDSF-treated cells (**Fig.**
 **7J**, and shown below)

However, there was subtle difference in TLR9-UNC93B1 binding affinity when we
 compared WT with PPT1-deficient cells (data not shown), suggesting that PPT1 may
 not be the only depalmitoylation enzyme involved in this process. Therefore, we think
 that HDSF may inhibit other palmitoyl thioesterases and have a broader target besides
 PPT1. Previous literature had shown that Hexadecylfluorophosphonate (HDFP), which
 has a similar structure with HDSF, could inhibit many serine hydrolases including PPT1

(see below).

 Modified from *Martin BR, Wang C, Adibekian A, Tully SE, Cravatt BF. Global profiling of*
 *dynamic protein palmitoylation. Nat Methods. 2011 Nov 6;9(1):84-9. doi:*
 *10.1038/nmeth.1769. PMID: 22056678; PMCID: PMC3248616.*

 Thus far, only 5 lysosomal enzymes were known to depalmitoylate proteins. We
 screened PPT1, ABHD17a, and ABHD17b. It is also possible that other unidentified
 thioesterases could also depalmitoylate TLR9. Accordingly, we have softened our
 previous conclusion that PPT1 was only depalmitoylation enzyme in the results (**p.14**)
 and discussed this result in the discussion (**p.18**).

 In summary, TLR9 palmitoylation regulates TLR9 trafficking from the Golgi to early
 endosomes, while TLR9 depalmitoylation controls the release of TLR9 from UNC93B1
 in endosomes. Inhibition of either step would result in less free cleaved TLR9 in
 endosomes and therefore less signaling.

 Regarding TLR9-Mut2:

 Yes, that is a sharp observation. There were no such weakened cleavage bands in our

DHHC3 (Fig. 5C, 7D, S7A, input panels) or PPT1-deficient cells (Fig. 6F,7F, S7A,
 input panels). However, there is weaker cleavage bands in TLR9-Mut2 cells in the input
 panel, before CpG-Bio pulldown (Fig. S7A and shown below).

We also performed cellular fractionation experiments on Mut2 cells, mTLR9^{Mut2} had no
 such trafficking defect, suggesting that its mechanism may be partially different (Fig.
 S7B, compared to Fig. 7I, and shown below).

This result indicated that the TLR9-Mut2 phenotypes are not entirely due to the loss of
 palmitoylation. One possible explanation could be that these two mutations play a role
 in TLR9 cleavage process itself. Multiple lysosomal proteases are required for TLR9
 cleavage. These two mutations may result in an altered structure, or may interfere with
 this enzymatic process. Given that TLR9 is a highly conserved protein, these two sites
 may have additional functions beyond palmitoylation. We have included this
 discrepancy in the results (p.15) and discussion (p.17).

*Perhaps some quantification of cleaved-to-full-length ratio could be provided for the*
 *input, as well as for the CpG-bound fraction, in 7B (and/or 4E). It looks like cleavage*
 *is efficient in both the Zddhc3 and Ppt1 knockouts (Fig 7D, F inputs) but it is hard to*
 *be sure from the blots.*

We apologize for not providing quantification. Upon reanalysis of the input panel, we
 think that the loss of DHHC3 did not affect cleavage efficiency of mTLR9. We have
 updated these statistics in Fig. S7A.

The loss of PPT1 did not affect cleavage efficiency of mTLR9 either. We have updated

these statistics in **Fig. S7A**.

3) I think the BDCA2-DTR mixed chimera experiment is not explained clearly. It could
 help if in supplemental figure 3C the top donor was labelled as "CD45.1+ WT or
 CD45.1+ *Ppt1*^{-/-}" which I think must be the design here. Under the conditions of this
 experiment, after DT depleting the BDCA2-DTR *Ppt*^{+/+} cells, still half of all other
 hematopoietic cells will be *Ppt*^{-/-} from the CD45.1 donor. It is not, as stated in the text,
 that "only pDCs possessed PPT1 deficiency," rather, you are depleting the WT pDC
 from the BDCA2-DTR donor, leaving only PPT1-deficient pDC but also half of
 everything else PPT-deficient. While it is likely that the pDC are the source of the
 difference in serum IFN α , I don't think that can actually be concluded from this
 experiment. To control for this, you could have included controls such as 100%
 BDCA2-DTR PPT1^{+/+} and 100% BDCA2-DTR PPT1^{-/-}; when treated with DT, pDC
 would be eliminated completely and the remaining hematopoietic cells would be 100%
 WT or 100% PPT1^{-/-}. If there were any differences between those two groups, it could
 point to a non-pDC source of PPT1-dependent IFN α .

We agree that the mixed chimeras have many caveats, and could only draw limited
 conclusions. Your suggestion is of course the best desired method, though it would
 take us at least 9 months to cross twice and make chimeras.

The mixed chimera approach is generally accepted in pDC studies due to the lack of
 a good pDC-specific Cre line (see review below). The main reason is that, though half
 of everything else is PPT1-deficient, the other PPT1-sufficient half compensates for it,
 thus making the whole hemopoietic environment (except pDCs) somewhat wildtype.

Strain	Cells Depleted	Caveats
Igax -DTR (Jung et al., 2002)	cDCs, metallophilic, marginal zone, sinusoidal, and alveolar macrophages, activated CD8 T cells, plasma cells	Repeated DT treatment leads to death, requires Igax -DTR into WT BM chimeras No depletion of pDCs
Igax -DOG (Hochweiler et al., 2008)	cDCs, splenic macrophages	Depletion of other cell types not evaluated
Zbtb46 -DTR (Meredith et al., 2012)	cDCs	DT treatment leads to death, requires zDC-DTR into WT BM chimeras
Zbtb46 -LSL-DTR (Loschko et al., 2016a)	cDCs	Requires crossing to Cre strain active in cDC lineage
Cd207 -DTR (Knockin) (Kissenpfennig et al., 2006)	Langerhans cells, cDC1s in dermis and skin-draining LNs	
Cd207 -DTR (BAC) (Bennett et al., 2005)	Langerhans cells, cDC1s in dermis and skin-draining LNs	
CD207 -DTA (Kaplan et al., 2005)	Langerhans cells, CD103 ⁺ CD11b ⁺ cDC2s in small intestine	
Ly75 -DTR (Fukaya et al., 2012)	cDC1s, ~15% of splenic B cells	DT treatment leads to death, requires Ly75 -DTR into WT BM chimeras Depletion of germinal center B cells, migratory cDCs, and Langerhans cells untested but possible
Clec9a -DTR (Piva et al., 2012)	cDC1s, ~50% of pDCs	
Xcr1 -DTRvenus (Yamazaki et al., 2013)	cDC1s	
Karma (a530099) f9nk -DTR (Alexandre et al., 2016)	cDC1s	
Clec4e4 -DTR (Muzaki et al., 2016)	cDC2s and CD64 ⁺ CD3CR1 ⁺ macrophages in intestinal lamina propria and mesenteric LNs	DC depletion in other organs not analyzed
Mgl2 -DTR (Kumamoto et al., 2013)	Mgl2 ⁺ cDC2s in dermis and skin-draining LNs	
CLEC4C -DTR (Swiecki et al., 2010)	pDCs	
Siglech -DTR (Knockin) (Takagi et al., 2011)	pDCs	Loss of Siglec-H expression in homozygous knockin mice. Depletion of marginal zone macrophages untested but possible
Siglech -DTR (BAC) (Piva et al., 2012)	pDCs, marginal zone macrophages	
Cx3cr1 -LSL-DTR Igax -cre (Diehl et al., 2013)	CD103 ⁺ CD11b ⁺ CD3CR1 ⁺ cDC2s, CD11b ⁺ CD3CR1 ⁺ macrophages	Requires crossing to Cre strain active in cDCs
MM -DTR (Schreiber et al., 2013)	Monocytes and monocyte-derived macrophages	

 *Table 1. Functions of Murine Dendritic Cells. Durai V, Murphy KM.*
 *Immunity. 2016 Oct 18;45(4):719-736. doi: 10.1016/j.immuni.2016.10.010.*

Use of genetic knockout mice in mixed chimeras are also common, especially in the
 field of ILC2, whereas there is no proper conditional cre line. In our previously
 published work in cDC1s, we also used a similar mixed chimera approach for specific
 cDC1-deficiency using *Batf3*-KO (which is deficient in cDC1s) and PPT1 WT/KO (see
 below).

 *Figure S1F Schematics of the generation of Ppt1+/+:Batf3-/- or Ppt1-/-:Batf3-/- mixed*
 *chimeras. Ou et. al. J Exp Med. 2019 Sep 2;216(9):2091-2112.*

Lastly, pDCs are not the only cells possessing phenotype, thus making point of pDC-
 specific deletion less meaningful. We demonstrated previously that macrophages had
 a similar phenotype as pDCs (**Fig. 3E**). Insight from another reviewer also implicated
 B cells (**Fig. 3E** and **S3H**). Therefore, we have moved the chimera data to

supplemental figures (**Fig. S3G**), and changed the description in results (**p.8**) and
 discussion accordingly (**p.20-21**).

*Minor point:*

*1) I wonder if the labeling of murine DHHCs screened in Figure 5A is quite correct? I*
 *think there is not a mammalian Zddhc10 gene (and indeed in this paper's methods*
 *describing cloning of the mouse DHHCs this one is skipped) while Zddhc23 seems to*
 *be omitted from the figure but is described in the methods.*

Thank you for noticing this discrepancy. This is not a typo but a historical quirk in the
 naming. The protein number corresponds to the gene number with the exception of
 DHHC10/Zdhhc11, DHHC11/Zdhhc23, DHHC13/Zdhhc24, DHHC22/Zdhhc13 and
 DHHC24/Zdhhc22. Please refer to the summary table below for details.

Table 2. Mammalian DHHC Enzymes

DHHC enzyme [gene symbol]	cellular localization	size (no. of a.a.)	Uniprot	disease association
DHHC1 [zDHHC1]	ER, extracellular vesicular exosome	485	Q8WFX9	
DHHC2 [zDHHC2]	plasma membrane, recycling endosome membrane	367	Q9UIJ5	lymph node metastasis and independently predicts an unfavorable prognosis in gastric adenocarcinoma patients; colorectal cancer ⁷⁵
DHHC3 [zDHHC3]	Golgi	327	Q9NYG2	
DHHC4 [zDHHC4]	Golgi, ER	344	Q9NPG8	
DHHC5 [zDHHC5]	plasma membrane, dendrite	715	Q9C0B5	postsynaptic function affecting learning and memory ⁹⁶
DHHC6 [zDHHC6]	ER	413	Q9H6R6	
DHHC7 [zDHHC7]	Golgi	308	Q9NXF8	
DHHC8 [zDHHC8]	Golgi, cytoplasmic vesicle, mitochondrion	765	Q9ULC8	defects in the encoding gene may be linked to susceptibility to schizophrenia; synaptic regulation ZDHHC8 knockdown enhances radio-sensitivity and suppresses tumor growth in a mesothelioma mouse model
DHHC9 [zDHHC9]	Golgi, ER, cytoplasm	364	Q9Y397	mutations in the gene are associated with X-linked mental retardation
DHHC10 [zDHHC11]	ER	412	Q9H8X9	gain of ZDHHC11 gene may be a potential biomarker for bladder cancer and nonsmall cell lung cancer
DHHC11 [zDHHC23]	plasma membrane	409	Q8IYP9	
DHHC12 [zDHHC12]	ER, Golgi	267	Q96GR4	alzheimer's disease ⁴¹⁵
DHHC13 [zDHHC24]	ER ⁴¹⁶	284	Q6UX98	

Table 2. continued

DHHC enzyme [gene symbol]	cellular localization	size (no. of a.a.)	Uniprot	disease association
DHHC14 [zDHHC14]	ER ⁹⁷	488	Q8IZN3	deletion may be linked to development delay, ⁴¹⁷ activation through chromosomal translocation in patients with acute biphenotypic leukemia ⁴¹⁸
DHHC15 [zDHHC15]	Golgi	337	Q96MV8	mutations in the gene cause X-linked mental retardation type 91 ⁴²¹
DHHC16 [zDHHC16]	ER, cytoplasm ⁴²³	377	Q969W1	
DHHC17 [zDHHC17]	Golgi, Golgi-associated vesicle membrane, cytoplasmic vesicle membranes ⁴²⁶	632	Q8IUH5	memory and synaptic deficits in KO mice ⁴²⁷
DHHC18 [zDHHC18]	Golgi	388	Q9NUE0	
DHHC19 [zDHHC19]	ER	309	Q8WVZ1	
DHHC20 [zDHHC20]	plasma membrane	365	Q5W0Z9	
DHHC21 [zDHHC21]	Golgi, plasma membrane	265	Q8IVQ6	loss of protein function result in delayed hair shaft differentiation and hyperplasia of interfollicular epidermis and sebaceous glands
DHHC22 [zDHHC13]	Golgi, Golgi-associated vesicle membrane, ⁴⁸⁰ ER, intracellular membrane bound organelle	622	Q8IUH4	ZDHHC13 deficient mice develop alopecia, amyloidosis, and osteoporosis ⁴⁴¹ and have reduced bone mineral density ⁴⁴²
DHHC24 [zDHHC22]	Golgi, ER	263	Q8N966	Huntington's disease

*Jiang H, et al. Protein Lipidation: Occurrence, Mechanisms, Biological Functions, and*
 *Enabling Technologies. Chem Rev. 2018 Feb 14;118(3):919-988.*

REVIEWERS' COMMENTS

Reviewer #1 (expert in innate immunity and post-translational modifications):

From the revised version, I do see improved work on mechanism although it was still not very clear. I would suggest the authors to provide a scheme/cartoon, in which the proposed scientific findings can be concluded and shown.

Reviewer #2 (expert in SLE):

The authors have done an admirable job of responding to reviewer comments and adding additional experimental data, particularly related to TLR7.

While their data do indicate modulation of several immune alterations characteristic of SLE, it should be noted that they have only investigated the role of the palmitoylation and depalmitoylation pathways in one murine lupus model. In addition, their analysis of the impact of PPT1 deficiency on lupus nephritis does not include measurement of urine protein, the typical assessment of lupus nephritis activity. They should be cautious not to overinterpret the impact of their results on lupus pathogenesis as there are many mechanisms involved in that disease and the pathway being studied only impacts some of those mechanisms. They might slightly modify the title of the manuscript to indicate regulation of the TLR9 pathway rather than systemic lupus in general and should note limitations regarding the focus on only the one murine lupus system.

It is interesting that the standard drug for SLE, hydroxychloroquine, has been reported to inhibit PPT1. Moreover, analysis of the inhibitory effect of that drug on TLR9 vs. TLR7 pathways indicates more robust inhibition of the TLR9 pathway (Sacre K et al. Arthritis Res Ther. 2012), consistent with the authors' results documenting distinct requirements for PPT1 in depalmitoylation of TLR9 vs. TLR7. The authors emphasize another candidate therapeutic agent - HDSF - but in fact the most commonly used lupus therapeutic may already be the correct agent for inhibiting TLR9 signaling.

Reviewer #3 (expert in TLRs in lupus):

The authors have addressed all of my concerns to my satisfaction.

We would like to thank all reviewers for their constructive criticism of our manuscript. The changes are described below and also **highlighted** in the revision.

*Reviewer #1 (expert in innate immunity and post-translational modifications):
From the revised version, I do see improved work on mechanism although it was still not very clear. I would suggest the authors to provide a scheme/cartoon, in which the proposed scientific findings can be concluded and shown.*

Thank you for your suggestion. We have now included a working model in **Fig. S8E**.

*Reviewer #2 (expert in SLE):
The authors have done an admirable job of responding to reviewer comments and adding additional experimental data, particularly related to TLR7.*

While their data do indicate modulation of several immune alterations characteristic of SLE, it should be noted that they have only investigated the role of the palmitoylation and depalmitoylation pathways in one murine lupus model. In addition, their analysis of the impact of PPT1 deficiency on lupus nephritis does not include measurement of urine protein, the typical assessment of lupus nephritis activity. They should be cautious not to overinterpret the impact of their results on lupus pathogenesis as there are many mechanisms involved in that disease and the pathway being studied only impacts some of those mechanisms. They might slightly modify the title of the manuscript to indicate regulation of the TLR9 pathway rather than systemic lupus in general and should note limitations regarding the focus on only the one murine lupus system.

We agree with you that the claims on SLE were rather limited. We have changed the title to “Cyclical palmitoylation regulates TLR9 signalling and systemic autoimmunity in mice”. We have also deleted the claims about SLE in the abstract and introduction.

It is interesting that the standard drug for SLE, hydroxychloroquine, has been reported to inhibit PPT1. Moreover, analysis of the inhibitory effect of that drug on TLR9 vs. TLR7 pathways indicates more robust inhibition of the TLR9 pathway (Sacre K et al. Arthritis Res Ther. 2012), consistent with the authors' results documenting distinct requirements for PPT1 in depalmitoylation of TLR9 vs. TLR7. The authors emphasize another candidate therapeutic agent - HDSF - but in fact the most commonly used lupus therapeutic may already be the correct agent for inhibiting TLR9 signaling.

Yes, we absolutely agree. We have included this point and the citation in the discussion.

*Reviewer #3 (expert in TLRs in lupus):
The authors have addressed all of my concerns to my satisfaction.*

Thank you so much!